

# The origin of methane in the East Siberian Arctic Shelf unraveled with triple isotope analysis

[1,2]Célia J. Sapart*, [3,4]Natalia Shakhova, [3,4,5]Igor Semiletov, [1,6]Joachim Jansen, [7]Sönke Szidat, [5]Denis Kosmach, [5]Oleg Dudarev, [1]Carina van der Veen, [8]Matthias Egger, [9]Valentine Sergienko, [5]Anatoly Salyuk, [10]Vladimir Tumskoy, [2]Jean-Louis Tison and [1]Thomas Röckmann.

[1]Institute for Marine and Atmospheric research Utrecht (IMAU), Utrecht University, Princetonplein 5, 3584CC Utrecht, The Netherlands.

[2]Laboratoire de glaciologie, Universtié Libre de Bruxelles (ULB), Avenue Roosevelt 50, 1050 Brussels, Belgium.

[3]University Alaska Fairbanks, International Arctic Research Center, 930 Koyukuk Drive, Fairbanks, USA, 99775.

[4]Tomsk Polytechnic University, 30 Prospect Lenina, Tomsk, Russia.

[5]Russian Academy of Sciences, Far Eastern Branch, V.I. Il'ichov Pacific Ocenological Institute, 43 Baltiyskaya street, Vladivostok 690041.

[6]Department of Geological Sciences and Bolin Centre for Climate Research, Stockholm University, Frescativägen 8, SE 114 18, Stockholm, Sweden.

[7]Department of Chemistry and Biochemistry & Oeschger Centre for Climate Change Research, University of Bern, Freiestrasse 3, CH-3012 Bern, Switzerland.

[8]Department of Earth Sciences - Geochemistry, Utrecht University, Princetonplein 9, 3584CC Utrecht, The Netherlands.

[9]Russian Academy of Sciences, Far Eastern Branch, Institute of Chemistry, 159 Prospect 100-letiya Vladivostoka, Vladivostok 690022.

[10]Moscow State University, 1 Leninskie Gori, 119991, Moscow, Russia.

## Abstract

Methane ($CH_4$) is a strong greenhouse gas emitted by human activity and natural processes that are highly sensitive to climate change. The Arctic Ocean, especially the East Siberian Arctic Shelf (ESAS) overlays large areas of subsea permafrost that is degrading. The release of large amount of $CH_4$ originally stored or formed there could create a strong positive climate feedback. Large scale $CH_4$ super-saturation has been observed in the ESAS waters, pointing to leakages of $CH_4$ through the sea floor and possibly to the atmosphere, but the origin of this gas is still debated.

Here, we present $CH_4$ concentration and triple isotope data analyzed on gas extracted from sediment and water sampled over the shallow ESAS from 2007 to 2013. We find high concentrations (up to 500μM) of $CH_4$ in the pore water of the partially thawed subsea permafrost of this region. For all sediment cores, both hydrogen and carbon $CH_4$ isotope data reveal the predominant presence of $CH_4$ that is not of thermogenic/natural gas origin as



it has long been thought, but resultant from microbial $CH_4$ formation using as
primary substrate glacial water and old organic matter preserved in the
subsea permafrost or below. Radiocarbon data demonstrate that the $CH_4$
present in the ESAS sediment is of Pleistocene age or older, but a small
contribution of highly [14]C-enriched $CH_4$, from unknown origin, prohibits
precise age determination for one sediment core and in the water column.
Our data suggest that at locations where bubble plumes have been observed,
$CH_4$ can escape anaerobic oxidation in the surface sediment. $CH_4$ will then
rapidly migrate through the very shallow water column of the ESAS to escape
to the atmosphere generating a positive radiative feedback.
**1.Introduction**

The Arctic subsea permafrost harbors a very large active carbon pool
of similar size as the terrestrial Siberian permafrost reservoir (Shakhova et al.,
2010a). Between 12 and 5kyr Before Present (BP), the Holocene
transgression (Bauch et al, 2001) submerged extensive parts of the
Pleistocene age terrestrial permafrost in Northern Siberia, forming the very
shallow ESAS (Romanovskii et al., 2005). As a result, the formerly terrestrial
permafrost has been continuously exposed to increasing seawater
temperatures, salt and anoxic conditions (Dimitrenko et al., 2011, Nicolsky et
al., 2012) allowing the remobilization of carbon from the Pleistocene
reservoirs. The four key mechanisms controlling the release of Pleistocene
carbon from thawing subsea permafrost are gas hydrate degradation,
thermokarst development, the deepening of the permafrost active layer and
coastal erosion (e.g. Shakhova et al., 2005, 2009, 2010a,b, O'Connor et al.,
2010, James et al., 2016). Holocene age carbon originating mainly from
coastal erosion and riverine discharge (Charkin et al., 2011; Semiletov et al.,
2011, 2012; Karlsson et al., 2011, 2016) has accumulated on the ESAS shelf
and overlays the Pleistocene age sediment (Vonk et al., 2012, 2014 Feng et
al., 2013).
Under specific conditions and depending on its type and quality
(Schuur et al., 2013), the remobilized carbon can be used to produce $CH_4$, a
strong greenhouse gas (IPCC, 2013). Biogenic $CH_4$ is produced by
methanogenesis using as main substrates carbon dioxide ($CO_2$) and acetate
according to the following reactions:

(CO_2 reduction) $CO_2 + 4H_2 \rightarrow CH_4 + 2H_2O$

(Acetate fermentation) $CH_3CO_2^- + H_2O \rightarrow CH_4 + HCO_3^-$

In the deep Earth layers, thermogenic $CH_4$ can also be produced
abiogenically by thermal degradation of organic matter (Schoell, 1988) and
migrate towards the surface. A large part of the $CH_4$ formed in the seafloor is
removed by anaerobic oxidation with seawater sulfate in the surface
sediments (e.g. Reeburgh, 2007, Knittel and Boetius, 2009, Egger et al.,
2015) or in the water column where $CH_4$ can be consumed by aerobic



methanotrophic bacteria under specific nutrient and redox conditions (e.g.
Kessler et al., 2011, Mau et al., 2013, Steinle et al., 2015). Each type of $CH_4$
formation/removal pathway produces $CH_4$ with a characteristic isotopic
signature ($\delta^{13}C$ and $\delta D$) depending on the isotopic composition of the
substrate and the isotopic fractionation associated with the respective
chemical reaction involved. Hence analyzing the isotopic composition of $CH_4$
in the sediment or in the water helps to unravel its formation/removal
pathways (Whiticar, 1999, Quay et al., 1999).
At relatively high pressure and low temperature conditions in the
seafloor, e.g. within the ESAS shelf and slope, $CH_4$ molecules can be stored
as gas hydrate, i.e. encaged in frozen water molecules (e.g. Kvenvolden,
1988, Sloan et al., 2003, James et al., 2016). Below this gas hydrate stability
zone, $CH_4$ occurs as free gas and can be advected towards the surface
through faults in the sediment. Subsea permafrost has long been thought to
play the role of a gas quasi-impermeable cap, preventing $CH_4$ to escape from
seabed deposits (e.g. Soloviev et al., 1987, Bauch et al., 2001, Romanovskii
et al., 2005, Shakhova et al., 2010a, Dmitrenko et al., 2011, Nicolsky et al.,
2012). However, the warming of the ESAS bottom water, accelerated by the
current decline in sea-ice coverage and the presence of geothermic activity in
this region (Soloviev et al., 1987; Romanovskii et al., 2005) has largely
degraded the ESAS subsea permafrost. This may cause a destabilization of
the gas hydrate layers and provide more extensive pathways for gaseous and
dissolved $CH_4$ to escape from the sediment to the atmosphere, e.g. when
bubble plumes are released to the water column from the seabed (Shakhova
et al., 2014, 2015) (for further discussion on gas transport processes within
subsea permafrost see the SI Section 1). Shakhova et al., 2010b carried out a
multi-year survey to measure $CH_4$ concentrations in the ESAS waters and
showed that >80% of ESAS bottom waters and >50% of surface waters are
supersaturated with $CH_4$ with respect to the atmosphere. Measured
concentrations were anomalously high (up to 500 nM) compared to $CH_4$
values generally observed in oceans (~5 nM, Damm et al., 2008). Shakhova
et al. (2010b, 2014, 2015) further observed vigorous bubbling at high
concentration "hotspots" indicating that the water column $CH_4$ supersaturation
must result from a strong source in the sediment.
To disentangle the origin(s) of this $CH_4$ anomaly, we measured $CH_4$
concentration, stable isotope composition and (on selected samples)
radiocarbon content on sediment and water samples from several winter
campaigns and summer cruises from 2007 to 2013 on the ESAS shelf and
shelf edge.   While stable isotope analyses help identify the chemical
pathways involved in $CH_4$ removal and formation processes, radiocarbon
measurements give information on the age of the $CH_4$ substrate the
combination of the isotope information helps determine the possible origin(s)
of this gas.
**2.Method**
**2.1.Drilling and sediment sampling**

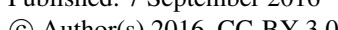



Summer surface sediment drilling and water sampling campaigns were
carried out from ships while the winter field campaigns were accomplished
using an equipment caravan, which traveled over the sea ice to the drilling
locations. In the latter case, casings were drilled through the fast ice into the
seabed, allowing dry drilling using a rotary drill with 4 m casing with a newly
built URB-4T drilling rig (made in 2011 by Vorovskii Factory for Drilling
Equipment, Ekaterinburg, Russia). Thawed and frozen sediments for each
core were subsampled straight after the drilling using ice screws for frozen
samples and a heavy plastic syringe-like sampler for thawed samples at 20
cm vertical resolution.
**2.2.     Gas extraction and measurement in sediments**
Sediment subsamples were immediately immersed in glass vials filled
with a saturated sodium chloride solution to drive gases out of solution and
capped with a septum for equilibrium in ultrasonic water bath at a temperature
of 20°C. $CH_4$ concentrations in the headspace were measured immediately
after equilibration with a SRI-8610c gas chromatograph, thermo-conductivity
detector, a helium detector, a flame-ionization detector and two columns:
molecular sieve 13X (6' x 1/8") and HayeSep D (6' x 1/8").  The amount of
gas in the vials was calculated from headspace concentrations, gas pressure
and solubility and the volume of liquid in the bottles following the method of
Wiesenburg and Guinasso (1979). $CH_4$ concentrations are reported relative to
sediment pore water volume, regardless of whether present as ice or water,
based on calculated total sediment water content, bulk density and dry
density (matrix or skeleton density) and sediment porosity. To estimate the
sediment porosity ($P_v$), we calculated the density of the sediment skeleton
($P_d$) and density of the sediment mineral particles ($P_s$), so that $P_v = (P_s-P_d)/P_s$;
where $P_d = P/(1+W_c)$; $P$ is the total density of the sediment sample, which was
calculated right upon sediment sample recovery by weighting the sample and
measuring its volume. $W_c$ is the sample moisture content (fraction of 1),
calculated as the difference between the weight of the wet sediment sample
and its dry weight. More details could be found in (Andersland and Ladanyi,
1994).

The stable isotope measurements were performed using a Continuous
Flow Isotope Ratio Mass Spectrometry (CF-IRMS) system as described in
Brass and Röckmann, 2010 and Sapart et al., 2011. Radiocarbon analyses
could be performed only on the largest samples. In that case, $CH_4$ was
preconcentrated and combusted to $CO_2$. The [14]C content of the $CO_2$ was
measured by accelerator mass spectrometry (Szidat et al., 2014) using a
specific gas inlet (Ruff et al., 2010).

**2.3. Gas extraction and measurement from seawater samples**

Water samples were collected directly from the Niskin bottles. Gas
from seawater samples was extracted using a modified headspace vacuum-



ultrasonic degassing method (Schmitt et al., 1991, Lammers et al., 1994). The
gas released was accumulated in an evacuated burette, to measure its
volume and is then expanded into a smaller flask for storage, and analysed
as described in Section 2.2.
**3.  Results and discussion**
We present results of $CH_4$ concentration, stable isotope composition
and (on selected samples) radiocarbon content on four shallow sediment
cores (<3m), four deep sediment cores (ID-11, IID-13, IIID-13, VD-13) (down
to 53m depth) and about fifty water samples from four coastal areas of the
ESAS: the Lena Delta (LD), the Buor-Khaya Bay (BKB), the Dmitry Laptev
Strait (DLS) and the Shelf Edge (SE) (Fig.1). All water and sediment sampling,
except for the ID-11 core, was performed at hotspot sites, i.e., at locations
where active gas bubbling from the seafloor and high concentrations of
dissolved $CH_4$ were observed as discussed in Shakhova et al., 2010a. The
location of core ID-11 is referred to as 'background site', where no gas
bubbling was observed. Here, the core was thawed all the way down to 53m
while the IID-13, IIID-13 and VD-13 cores were thawed down to 19, 17 and
12m, respectively. Note that sampling was continued through the deeper
frozen sediment for the three latter cores to 30, 50 and 35m respectively. For
more details on the lithology and sediment properties, see SI section 2 and
Fig.S1.
**3.2.  $CH_4$ in the "deep" sediment**
Depth profiles of $CH_4$ concentration, stable isotope composition ($\delta^{13}C$
and $\delta D$) and the radiocarbon content (in percent modern carbon, pmC) are
presented in Fig.2. In both hotspot and background cores, $CH_4$
concentrations are far above values observed in the water column and $CH_4$ is
strongly depleted in heavy isotopes in all sediment cores. $CH_4$ in the hotspot
cores IID-13, IIID-13 and VD-13 is more depleted in $\delta D$ and slightly more
enriched in $\delta^{13}C$ than in the background core. These differences can be
caused by the difference in lithology (Fig.S1), the heterogeneity in substrate
availability (Karlsson et al., 2011, 2016, Tesi et al., 2014, 2016) at the
different sampling locations, their distance from the coast and the depth of the
gas formation, which will be discussed in detail below.
The expected stable isotope signatures of the three potential $CH_4$
formation pathways in sediment (Whiticar, 1999): $CO_2$ reduction, acetate
fermentation and thermal degradation of organic matter are depicted together
with our water and sediment stable isotope data in a dual isotope plot (Fig.3).
Overall, the deep sediment core data (diamonds) fall in between the isotope
source signatures of the two biogenic pathways: carbonate reduction and
acetate fermentation. This could imply that $CH_4$ is formed by a mixture of both
sources. However, salinity measurements along the deep sediment cores
indicate the presence of interstitial seawater all the way down the cores.
When the seawater sulfate enters the marine sediment (Henrichs and
Reeburgh, 1987), it provides sulfate reducing bacteria with the electron



acceptor they need to outcompete methanogens for acetate (Lessner, 2009).
This indicates that "in-situ" (i.e. at the depth where the samples were taken)
acetoclastic $CH_4$ formation may be suppressed despite an abundance of
organic material. $CO_2$ remains therefore the most likely non-competitive
substrate for methanogens if "in-situ" formation would occur. Note that the
presence of sulfate along the core might be expected to involve anaerobic
oxidation of $CH_4$, but no significant enrichment in heavy isotopes in concert
with concentration decrease is observed in the stable isotope profiles of the
deep cores, except between 40 and 30m and between 4 and 0 m for the ID-
11 core. This is further discussed in section 3.4.
Stable isotope measurements further support the notion that $CH_4$ is not
produced at the sampling location using the infiltrated water, but migrates up
from deeper layers. The exceptionally low $\delta D$ values in the deep core
sediment could be explained by the use of isotopically depleted water as
substrate for $CH_4$ formation. Chanton et al, 2006 and Brosius et al., 2012
measured very low meltwater signatures ($\delta D(H_2O)$) of -135±25‰ and -
220±30‰, respectively) in old Arctic permafrost (Fig. 4). We suggest that
methanogens present in the thawing permafrost use and/or have used such
depleted permafrost meltwater, unfrozen porewater or water from the
hydraulic system (see SI, section 1) as a hydrogen source to form $CH_4$ with
low $\delta D$ values. Arctic modern seawater ($\delta D(H_2O)$ = -20‰ (Friedman et al.,
1964)), infiltrating the marine sediment from above, is too enriched to explain
the observed deuterium signature.
We conclude that the $CH_4$ present in the surface thawed subsea-
permafrost is formed mainly biogenically by reduction of $CO_2$ in deeper
(>53m) sediment layers where old, isotopically depleted meteoric (melt-)water
is still present, or/and by acetate fermentation occurring at depth where no
seawater sulfate is present. Our observations thus imply that $CH_4$ is not
formed "in situ" but that it migrates from a deeper reservoir to the surface of
the partially thawed ESAS subsea permafrost. This conclusion is supported
by the observation that the vertical profiles - especially the stable isotope
profiles - are relatively constant with depth. High $CH_4$ concentrations are also
observed in frozen sediment showing that gas can migrate through
permafrost even before it is completely thawed (see SI section 1 for further
discussion on this topic).
The $^{14}C$ of $CH_4$ from the hotspot cores covers a range from 0.79 to
3.4pmC corresponding to a radiocarbon age of 26 to 39kyBP. This indicates a
carbon substrate of Pleistocene age. For the ID-11 background core, $^{14}C$
values are unexpectedly high and vary from 87pmC (radiocarbon
age=1kyBP) to 2367pmC, which represents a substantial enrichment above
the natural background. The same applies for water samples from the SE.
Note that levels close to 100pmC indicate modern values. Even samples that
had been affected by the nuclear bomb testing in the 1950s and 1960s would
show levels below 200pmC and $^{14}C$ values >200pmC cannot be caused by a
known natural processes. As discussed in the SI section 3, local
anthropogenic nuclear contribution is the most likely explanation for these
elevated radiocarbon levels, however this contamination remains small in
comparison to the main $CH_4$ source as it can be observed on the Keeling plot



(Keeling, 1961)(Fig. S2). Whereas this $^{14}$C contamination complicates precise
age determination using radiocarbon (considering that $CH_4$ present in the
water is the result of a mixture of all possible sources), the $^{14}$C values
measured at the hotspot cores together with the very low $\delta D$ values give
strong evidences that $CH_4$ from old reservoirs (Pleistocene age or older) is
being released there.
**3.4.  $CH_4$ in the surface sediment**
The ID-11 background site was the only coring location where no
active bubbling was observed from the surface sediment. Here, the top 5.8m
consist of a thick silty-clay layer (Fig.S1) of marine origin as indicated by the
higher salinity and silica concentrations, typical of a marine environment
enriched in diatoms (Fig.5). The increase in sulfate concentration together
with the strong $CH_4$ concentration decrease and the isotopic enrichment in
both $^{13}$C and D towards the sediment surface indicate that most of the $CH_4$
diffusing through this thick Holocene marine layer is removed by anaerobic
oxidation with sulfate at the surface sediment before reaching the water. This
has been reported by Overduin et al., 2015 for a site close-by.
This surface layer may also act as a physical barrier preventing gas to
migrate towards the surface directly. The $CH_4$ concentration increase from 9
to 5.8m depth without strong isotopic shifts (Fig. 5) and the acoustic data (Fig.
S3) show that gas accumulates under this relatively impermeable layer. The
upward $CH_4$ flux is therefore highly restrained facilitating horizontal gas
transport towards e.g. open taliks or hotspot locations where part of the gas
can be released as bubbles to the water column without being oxidized. The
isotopic signatures of the $CH_4$ in the pore water of the hotspot cores do not
show isotopic fractionation toward the surface (Fig.2). This is likely because
at these sites, ebullition processes may physically disturb the sulfate reducing
layer and reduce the amount of $CH_4$ subject to anaerobic oxidation (only
dissolved $CH_4$ is accessible for methanotrophic organisms).
The shallow sediment samples have $^{14}$C values from 3 to 88pmC
(radiocarbon age= 1-26kyBP) showing the presence of old $CH_4$ in surface
sediment of relatively modern age and thus confirming the migration of old
gas from deeper reservoir towards the surface. Note that the overall low
content of organic carbon (<2.3%) with a high fraction of lignin (Bröder et al.,
2016; Tesi et al., 2015; Vonk et al., 2014) in the surface sediment (Fig.5)
would anyway inhibit $CH_4$ formation in the marine layer hence in situ
methanogenesis there is highly unlikely.
**3.5.  $CH_4$ in the water**
$CH_4$ in water samples is more enriched in heavy isotopes than in
sediment samples. The highest $CH_4$ concentrations in the water column are
observed close to the seabed and at the surface in the presence of sea ice
(Fig.2a blue triangles). The $^{14}$C values of water samples are between 83 and
9560pmC (radiocarbon age= 2kyBP to strongly enriched above natural





present day values) (Fig.2d) (SI section 3). For the water samples, we only
encountered the highly enriched $^{14}$C values at the shelf edge, but we suggest
that this signature is likely diluted over the shelf, where old $CH_4$ from the
sediment is added. This could explain the broad range of pmC values
observed in the water column.
Two scenarios may explain the difference in stable isotope signatures
between the water- and sediment samples. The first assumes a mixture of
depleted old $CH_4$ from $CO_2$ reduction from the sediment (as identified above)
with a source that is more enriched in heavy isotopes. This source could be
either "in-situ" production in the water or thermal degradation of organic
matter in the deep Earth layers. In the marine environment, $CH_4$ could in
principle be produced at the pycnocline, where natural differences of water
density create a "fluid bottom", on which organic particles and pellets could
accumulate as substrate for "in-situ" methanogenesis (Damm et al., 2008,
Karl et al., 2008, Sasakawa et al., 2008). In the ESAS, the pycnocline is very
shallow and a very low primary production is expected because of darkness
and ice cover in the winter and because of the little available sunlight in the
summer due to the high solar zenith angles and the very turbid waters (light
penetrates only down to 40cm). Water "in-situ" production of $CH_4$ is therefore
very unlikely. Thermogenic emissions from the sediment are possible,
especially from the fault zone near the shelf edge where we find strong heavy
isotope enrichment in the water. We have not measured any $CH_4$ with a
thermogenic isotopic signature in our deep sediment cores, but one should
note that deep sediment drilling in the shelf edge was not possible because of
the rough ice conditions.
As a second hypothesis, the isotopic signature of the water samples
may also result from substantial aerobic (e.g. in the water under the sea ice)
or anaerobic (in the surface sediment) oxidation of $CH_4$ emitted from the deep
sediment. To test this quantitatively, we plotted oxidation slopes (showing the
evolution of the isotopic signature of the remaining $CH_4$ after oxidation) for the
largest and lowest  fractionation factors ($\varepsilon_D$: 98-324‰ and $\varepsilon_{13C}$: 2-38‰) found
in the literature (Table 1) and using the samples with the highest
concentration as initial source in a dual isotope plot (Fig.4). At first sight and
when using these wide ranges of fractionation factors, all water samples may
be the result of $CH_4$ oxidation in the surface sediment or in the water. In such
a case, however, the samples with the more enriched isotope signatures
should correspond to the lower concentration, which is not the case,
especially for the shelf edge samples that show the opposite pattern (Fig.2,
yellow triangles). Oxidation alone, without the addition of $CH_4$ from another
source, can thus not explain the stable isotope difference between the water
and sediment samples.
$CH_4$ measured in the water samples is mainly dissolved $CH_4$, because
of the low probability to trap bubbles in the Niskin bottles during sampling. In
the sediment, gas bubbles have time to equilibrate with pore water, especially
when the gas is trapped under relatively impermeable sediment, e.g. the
Holocene marine silty-clay layer. Therefore, we assume that in the sediment,
the pore water is in equilibrium with the gas bubbles, while we suggest that in
the seawater bubbles travel too rapidly to reach an isotopic equilibrium with





the dissolved gas. This means that the $CH_4$ isotopic signature of the gas
bubbles may not strongly affect the $CH_4$ dissolved in seawater. Our triple
isotope observations therefore show that $CH_4$ dissolved in ESAS seawater
likely originates for a large part from old deep gas that is only partially
oxidized in the sediment surface. This gas mixes into a small background
reservoir that is highly enriched in $^{14}C$ and causes the higher pmC values in
the seawater in comparison with pore water sediment.
**4. Conclusion**
Our triple isotope dataset of $CH_4$ from the sediment and water of the
ESAS reveals the presence of large amounts of biogenic $CH_4$ in the shelves.
This gas is formed continuously from old substrates at depth and/or has been
stored as gas hydrate and/or gas pockets in or below the subsea permafrost.
We show that already today this $CH_4$ from deep, old reservoirs
migrates through the thawing permafrost towards the seafloor, either
dissolved in the sediment pore water or as gas bubbles, to reach the
seawater. The very shallow depth of the ESAS allows a short path for these
bubbles to reach the atmosphere, and the major sea ice extent decline in this
area may additionally enhance the escape to the atmosphere. No quantitative
estimate of this $CH_4$ source is to date possible, but a rise in temperature will
enhance microbial formation and permafrost thawing, hence emissions of
biogenic $CH_4$ from the deep subsea permafrost of the ESAS are expected to
play an increasingly important role for the radiative forcing of the Earth in the
future.
Variations in $CH_4$ isotopic signatures in air trapped in polar ice cores
have been studied to investigate the cause(s) of the $CH_4$ increase observed
during past warming events (Sowers, 2006, Fischer et al., 2008, Bock et al.,
2010). The authors measured a shift towards lighter $CH_4$ stable isotope
values together with a temperature increase and they concluded that a rise in
wetland $CH_4$ emissions is the most likely explanation. Our results show that
thawing subsea permafrost emits large amounts of $CH_4$ that is depleted in
heavy isotopes and that such emissions cannot be easily distinguished from
Arctic wetland emissions when looking only at stable isotope data.
Considering the sensitivity of subsea permafrost to warming, such emissions
may have played a significant role in the large $CH_4$ rises recorded in ice core
air during warming events in the past.

**ACKNOWLEDGEMENTS**
We are grateful to the help of Gary Salazar (University of Bern) with the $^{14}C$
measurements. This research was supported by the Russian Government
(No. 14.Z50.31.0012/03.19.2014), the US National Science Foundation (OPP
ARC-1023281;   0909546);   the   NOAA   Climate   Program   office
(NA08OAR4600758).  N.S., D.K., and O.D. acknowledge support from the
Russian Science Foundation (No. 15-17-20032).



We would like to thank Jorien Vonk, Alain Prinzhofer, Helge Niemann, Nadine Mattielli and Dominique Weiss for the fruitful discussions and precious help in the interpretation of these data and Rebecca Fisher, Elise van Winden and Joralf Quist for their help with the stable isotope measurements and system calibration.

**AUTHOR CONTRIBUTION**

**C.J.S., N.S., T.R., J.J., S.S., I.S., J.L.T. and M.E. worked on the scientific interpretation and wrote the manuscript. N.S. and I.S. planned the research and organized the multiyear fieldwork campaigns. C.vd.V., C.J.S., S.S. and J.J. performed the isotopic analyses. I.S., D.K., O.D., V.S., A.S. and V.T. performed the water sampling, sediment drilling, the headspace preparation and CH$_4$ concentration measurements on the field.**

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





**Table and Figures**

| Processes | $\varepsilon$ ($^{13}$C) (‰) | $\varepsilon$ (D) (‰) |
|---|---|---|
| **Anaerobic Methane Oxidation** | | |
| Marine sediment and water column (Whiticar and Faber, 1986) | 2 - 14 | |
| Marine sediment – Baltic Sea (Martens et al., 1999) | 11 - 13 | 100 - 140 |
| Brackish/marine sediment – Bothnian Sea (Egger et al., 2015) | 9 | 98 |
| Marine sediments – Alaska (Alperin et al., 1988) | 8 - 10 | 134 - 180 |
| Marine sediment - Hydrate Ridge, Pacific Ocean (Holler et al., 2009) | 12 | 105 - 156 |
| Marine sediment - Mud Volcano, Mediterranean Sea (Holler et al., 2009) | 19 - 23 | 139 - 185 |
| Microbial mat – Black Sea (Holler et al., 2009) | 34 - 38 | 273 - 324 |
| Water column – Black Sea (Reeburgh et al. 2006) | 16 - 24 | |
| Water column – Black Sea, Cariacco Basin (Kessler et al., 2006) | 20 - 22 | 181 - 221 |
| Incubation, nitrite-driven AOM (Rasigraf et al., 2012) | 27 - 32 | 272 - 317 |
| **Aerobic Methane Oxidation** | | |
| Seep field offshore CA, USA (Kinnaman et al., 2007) | 22 - 30 | 156 - 320 |
| Laboratory incubation (Coleman et al., 1981) | 13 - 25 | 97 - 350 |
| Laboratory incubation (Feisthauer et al., 2011) | 15 - 28 | 110 - 232 |

Table 1: Literature review of the fractionation factor $\varepsilon$ associated with CH$_4$ oxidation in the marine environment.






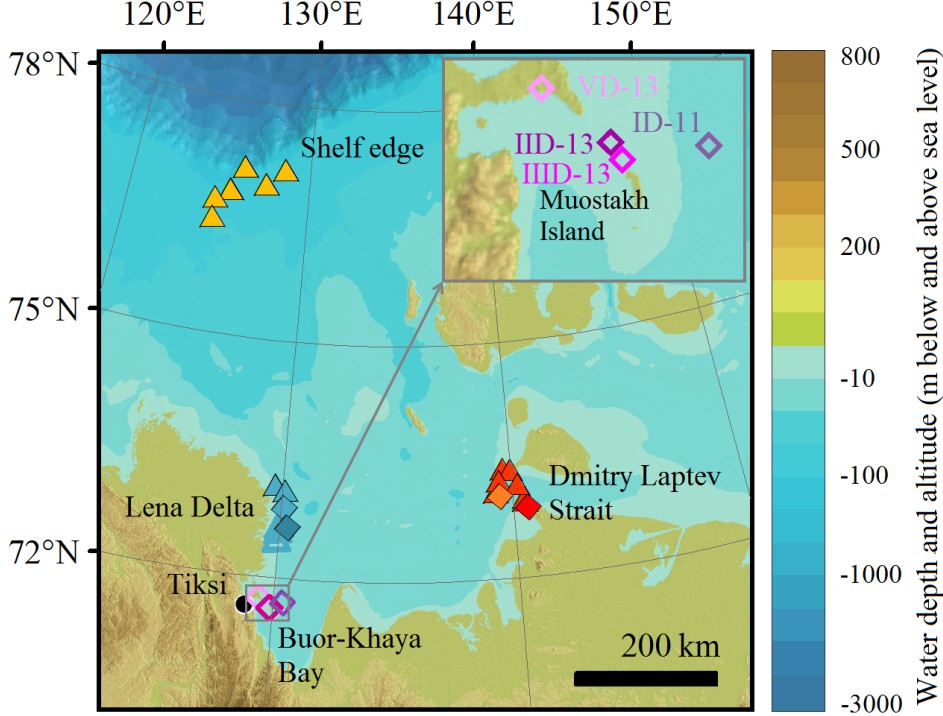


**Figure 1: Sampling location. Water sampling (triangles), sediment drilling (diamonds). Summer sampling (close symbols) and winter sampling (open symbols). The color legends of the deep sediment cores are shown on the top right.**




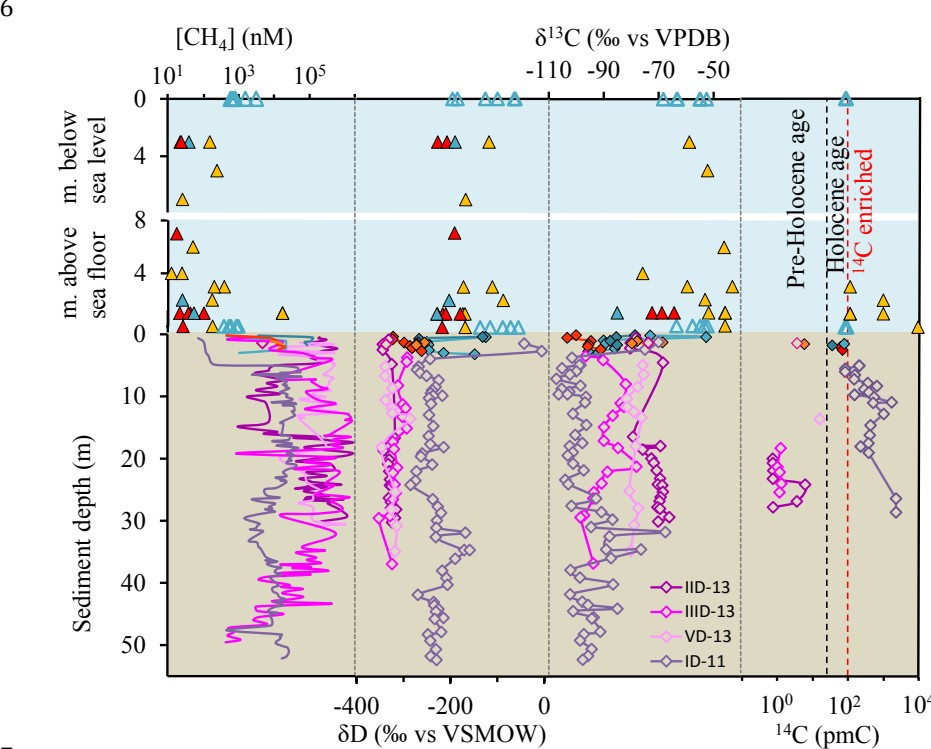

**Figure 2: CH₄ data from sediment and overlying water sampled at the East Siberian Arctic Shelf.**
**Water sampling (triangles), sediment cores (diamonds). Summer sampling (close symbols) and**
**winter sampling (open symbols). Buor Khaya Bay (purple, ID-11: background site and IID-13, IIID-**
**13 and VD13 hotspot sites), Dmitry Laptev Strait (red and orange), Lena Delta (light blue) and Shelf**
**Edge (yellow) (see Fig.1 for detailed location). (a) CH₄ concentrations, (b) δD (‰ vs VSMOW), (c)**
**δ¹³C (‰ vs VPDB), (d) ¹⁴C (pmC). The red dotted line corresponds to modern values (i.e., 100pmC)**
**and the black dashed line corresponds to the onset of the Holocene (11,000 years BP). Note that y-**
**axis for the water samples is divided in two sections. The upper part corresponds to the depth from**
**the sea surface and the lower part corresponds to the depth from the seabed.**





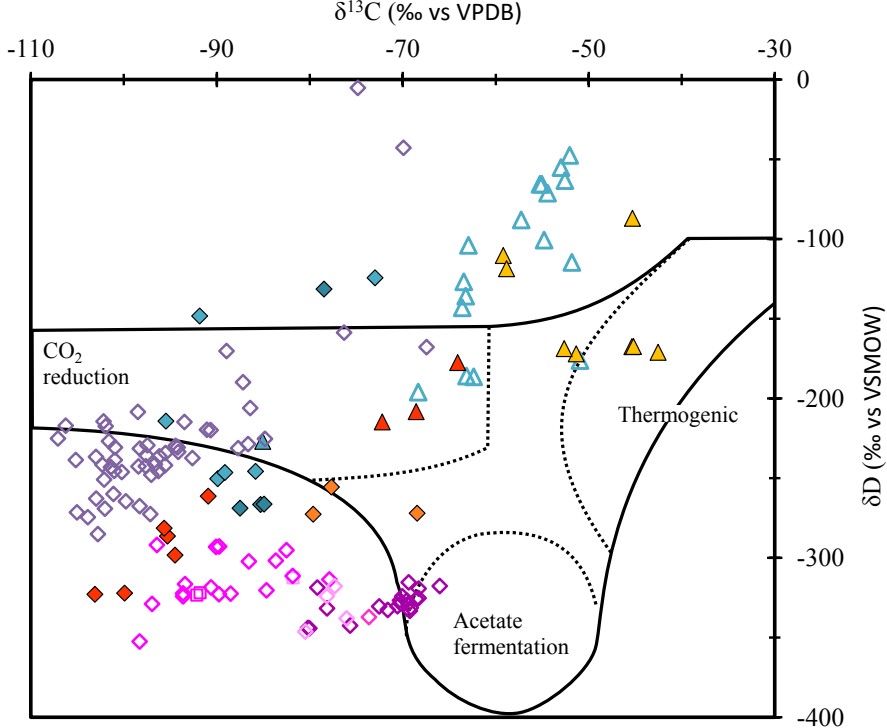


**Figure 3: Dual-isotope CH₄ plot. Legend is similar to Fig.2. Squares correspond to sample extracted directly from porewater in the sediment. Areas delimited by black lines correspond to the three main CH₄ formation processes and their isotopic signatures are retrieved from Whiticar, 1999.**













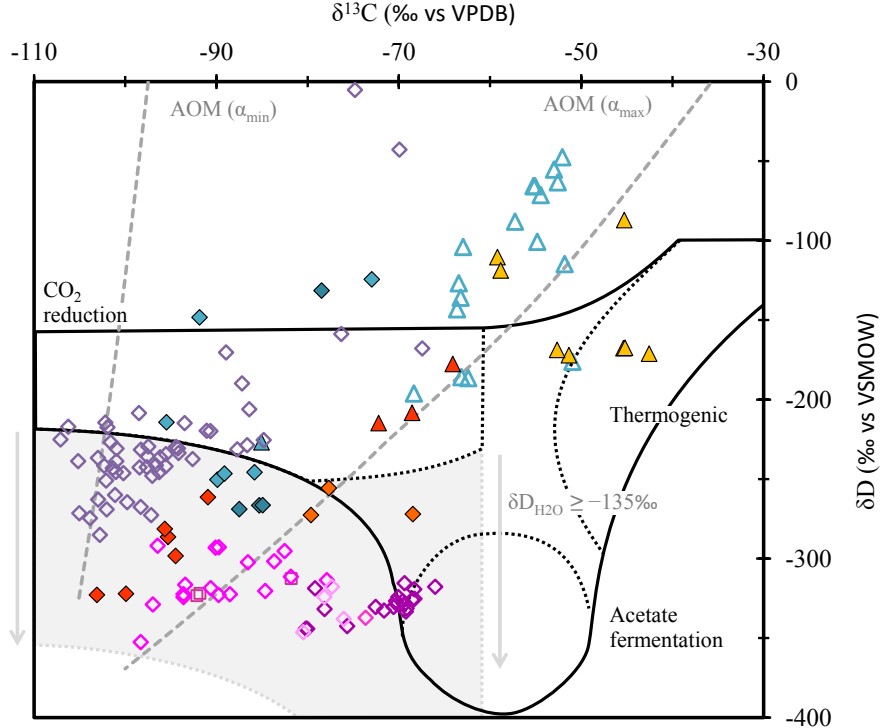

**Figure 4: Dual isotope plot for the CH₄ samples from the ESAS area. Water sampling (triangles), sediment cores (diamonds). Summer sampling (close symbols) and winter sampling (open symbols). Buor Khaya Bay (purple, ID-11: background site and IID-13, IIID-13 and VD13 hotspot sites), Dmitry Laptev Strait (red and orange), Lena Delta (light blue) and Shelf Edge (yellow) (see Fig.1 for detailed locations). Squares correspond to sample extracted directly from porewater in the sediment. Areas delimited by black lines correspond to the three main CH₄ formation processes and their isotopic signatures are retrieved from Whiticar, 1999. The shaded grey zone represents the possible CH₄ isotope signature associated to CO₂ reduction using old glacial water as substrate. Grey dashed lines indicate oxidation slopes.**

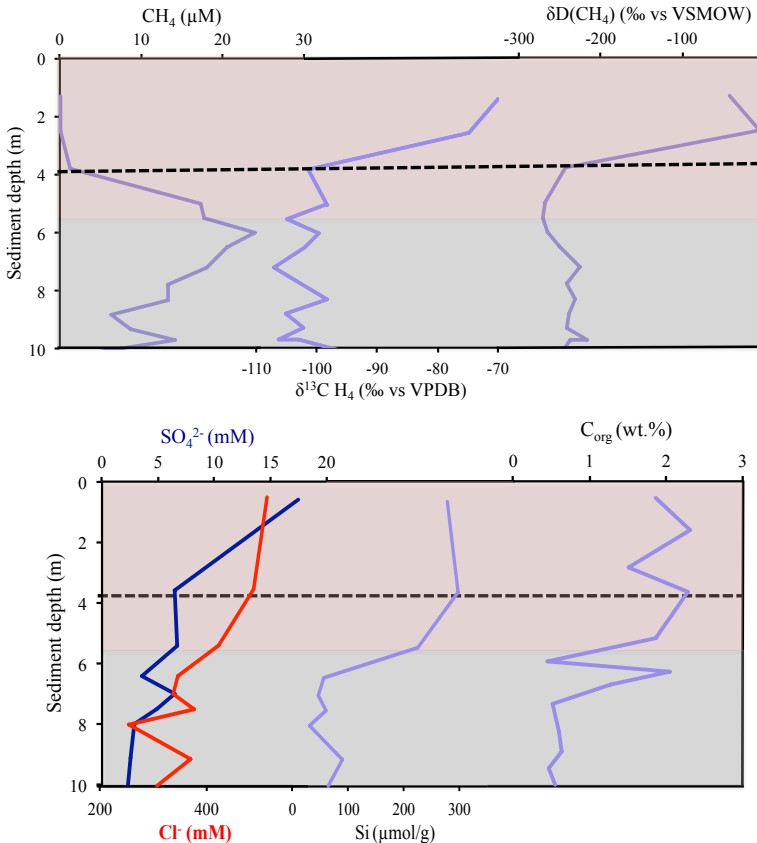

**Figure 5: Close-up of the CH₄ concentration, stable isotope and other biogeochemical data of the**
**surface of the background sediment core ID-11. Red shaded area corresponds to the marine**
**sediment deposited during the Holocene transgression and the grey shaded area corresponds to the**
**thawed permafrost layer. The black dotted line corresponds to the depth where CH₄ oxidation starts**
**to occur.**