# Peer review of "The origin of methane in the East Siberian Arctic Shelf unraveled with triple isotope analysis"

_Biogeosciences, 2016_

## Referee Comment (RC1) · Anonymous Referee #1 · 19 Oct 2016

The manuscript (ms) by Sapart et al. presents valuable information on the origin of methane in the East Siberian Arctic Shelf (ESAS), long been discussed based on assumptions and speculations. I like that this data set represents results of triple-isotope analysis. To my knowledge, this is first presentation of such a kind collected in the marine Arctic. This fact itself makes this ms novel. Besides, the data are from the potentially globally important region, because a predominant fraction of submarine permafrost is there; the ESAS was suggested to incorporate much of organic carbon and, probably, hydrocarbon stocks and hydrates in the sediments beneath permafrost. It was also shown to be warming due to natural warming as well as global warming. I believe this new data would be of great interest to scientists working in different disciplines and areas of the Arctic: to geologists, bio-geochemists, oceanographers, atmospheric scientists, climatologists and climate scientists.

[Figure]

Authors report methane concentrations and triple isotope data analyzed on gas extracted from sediment (four long cores down to ~53m depth) and water sampled over the ESAS from 2007 to 2013. Samples were taken from different areas of the ESAS, which represent different state of subsea permafrost (frozen to thaw) extending from the coastal zone to the outer shelf. The manuscript is clearly written. The approach is valid and the applied techniques are appropriate – I have no questions regarding triple isotope measurements accomplished by the authors in the best European laboratories. I can't completely agree with the authors that that the predominant presence of methane is not of thermogenic/natural gas origin, but resultant from microbial methane formation using as primary substrate glacial water and old carbon preserved in the subsea permafrost or below, but I can share the logic of the authors of this ms, which is mostly resultant from limitations possessed by currently available methods of triple-isotope data interpretation. It is interesting that most of methane in the ESAS sediments is of Pleistocene age or older – in my understanding, this should mean that submarine permafrost is somehow permeable for gases. I am not satisfied with explanations presented by the authors regarding the contribution of super-modern radiocarbon in methane – anthropogenic sources would have been explaining this fact plausibly if it has been methane from the water column, not from the sediment cores. I understand that it must be very complicated to come up with more realistic explanations, but the authors should be working further on it.

I can raise several questions regarding this work. Specifically: - why did not authors consider results published by Franke and Cramer (2005) and Bussmann et al., (2013), who presented clearly thermogenic signature of methane sampled in the same area? -Why did not authors collected gas from bubbles included in the sea ice (reported by Shakhova et al., 2010) to measure triple-isotope signature of methane? Would not it give a kind of integral isotope signature of methane potentially approaching the atmosphere? -Why did not authors consider release of super-modern methane found further off-shelf (lines 327) by groundwater transported from the land (lines 327-329)? - How would the authors explain enrichment of d13C by <50 per mil if residence time

of shelf water is only few months?

Many other questions could be raised, but I realize that this manuscript is based on multi-year work in the harsh Arctic environment. It is clear to me, that one paper, even incorporating that extensive data set, cannot answer all scientific questions regarding the complex, and previously unstudied, Arctic marine methane cycle. I appreciate that the authors have been accumulating data for a long period trying to cover as much aspects of this novel topic as possible. I also understand limitations possessed by current state of isotope biogeochemistry, which make it difficult to interpret isotope data collected in actual environmental conditions where methane of different origin, age, from different sources could be contributing differently in different areas – it differs so much from all idealized conceptions used to interpret the data. I suggest that these questions would be addressed in further work on this topic and the current ms would be taken as a baseline, relative to which results of further investigations in this area could be evaluated. At this point, I recommend that this paper is published as is.

―――――――――――――――――

---

## Referee Comment (RC2) · Anonymous Referee #2 · 24 Oct 2016

Overall:

The paper: "The origin of methane in the East Siberian Shelf unraveled by triple isotopic analysis" reports isotopic data from methane in sediments and seawater. This unique data set comprises methane $\delta^{13}C$, $^{14}C$ and $\delta D$ values with concentrations enabling an unusual insight into isotopic shifts between sediments and sea water but also between different sediment cores in this polar region. The data clearly reveal the predominance of biogenic methane. Long term speculations about thermogenic/natural gas methane sources could not be confirmed. Beside this basic news the most surprising outcome is the fact that methane in sediments of the ESAS shelf is much more depleted in $^{13}C$ and D than expected when considering the two main pathways of methane formation, i.e. acetate fermentation and CO2 reduction. Unfortunately the interpretation of this unique data set is biased by just focusing to prove the existence of methane diffusing from larger depths by thawing permafrost to the sediment surface and further through the water column up to the atmosphere.

There is no doubt about the existence of carbon sources from Pleistocene age. Old particulate organic matter derived from permafrost soils, ice complexes and coastal erosion and transported by the Lena represents an important contribution to Siberian shelf sediments. Hence the option of a more recent methane formation with old C (terrestrial C?) should be at least also discussed.

Pleistocene aged water is argued to be involved in methane formation by CO2 reduction. While I am able to follow that argument of upward diffusing of Pleistocene aged methane as one possibility, the data are ambiguous and should be discussed in a broader context. Especially as both the $\delta^{13}C$ and the $\delta D$ values are clearly out of the range of bacterial methane formation by both pathways, i.e. CO2 reduction and acetate formation. The samples shown here have either $\delta D$ values or $\delta^{13}C$ values untypical for the proposed pathways. The shift in $\delta D$ values from hotspot cores to the background core is conspicuous. I would appreciate a discussion of this circumstance.

Furthermore, the hotspot sediment cores show $\delta D$ values, which are in a "normal" range for acetate formation, while just the $\delta^{13}C$ values vary in a broader range. However, this pathway is unable to prove inclusions of old glacial water in methane. In contrast, the background core rather refers to CO2 reduction as pathway of methane formation but this core is not localized at a "hotspot" position and also not Pleistocene aged. Especially in the context of the hydrogen sources in methane, further non-competitive pathways are worthwhile to be discussed. Summarized, it remains questionable if the isotopic signature of methane (especially the $\delta D$ values as evidence for the CO2 reduction pathway) in the sediments is

really 1:1 related to the former pathway of methane formation or if there are additional fractionation effects which modify the final signature.

Concerning the sea water data, the main result is to see the decoupling between sediments and water above. Without any information on the bathymetric regime any interpretation remains speculative.

Although I completely agree that this data set raises much more new questions than it is able to answer, a careful interpretation of this data set would improve the quality of the discussion. Finally, it would be worthwhile to revise the conclusions and those should be based on the data rather than on speculations which remain to be proven at this stage. I recommend major revisions.

Details:

Abstract

Line 30-37 is an introduction only remotely related to the data discussed here

Line 49-53 this assumption is not proved by the data shown here

Introduction

In general a short introduction in using $\delta^{13}C$ and $\delta D$ values is needed

Processes which might modify the signature additionally to pathways of formation are completely missing and should be included

Line 65- concerning the remobilization of carbon from the Pleistocene: terrestrial carbon transported by the Lena River into the shelf sediments should also be mentioned, see: Winterfeld et. al., 2015 Biogeoscience)

Line 78 Further pathways for methanogenesis should be at least mentioned (methylotrophic with non-competitive substrates- see also Whiticar et al, 1999)

Line 91-96 The "Whiticar scheme" considering the relationship between the isotopic composition and pathway of methane formation has been developed for sediments. Using this scheme for sea water data should include at least a discussion about further fractionation effects in order to avoid over-interpretation of data. For example: Methane diffusion through sediments might induce fractionation effects just described by Prinzhover and Pernaton 1997. Further, there is no proof that the isotopic signature of methane dissolved in sea water

(outside a plume) can be used in a simple way for source identification of methane formation in sediments or from gas fields

Line 97-122 gives a detailed description about potential methane source at the ESAS, however, that paragraph about potential gas hydrates and gas bubbles in that region does not introduce the topic of the paper and is not a helpful tool to understand the isotopic data shown here. Further, this paragraph is misleading as the data shown in the paper are not in that range to push forward the knowledge about the role of gas hydrates and gas bubbles and are not needed to introduce the paper.

Line 115 this citation is wrong in that context. Measured concentrations should be related to the atmospheric equilibrium concentration in ESAS

I recommend adding supplement information into the introduction (physical factors) as this contribution is essential for understanding and interpretation of this data set

Methods

Line 173 the term "largest samples" is misleading and should be improved

Results and Discussion
For my knowledge these is the first data set of methane isotopic signatures in frozen sediments hence potential freezing effects on isotopic fractionation should at least be taken into account
Line 213-215 how differences in lithology influence differences in the isotopic signature should be discussed in the main text

Line 219-222 most of the isotopic signatures of the deep sediment cores are not included in the range of CO2 reduction or acetate formation. Potential reasons should be discussed.

Line 223and Figs 3 and 4: yes, methane is unusually depleted in $\delta^{13}C$ and $\delta D$. The samples shown here have either $\delta D$ values or $\delta^{13}C$ values untypical for the proposed pathways. More in detail: for acetate formation the $\delta^{13}C$ values are untypical while the $\delta D$ are in a "normal"

range. However, this pathway is unable to prove the inclusion of old glacial water. But these cores are from the "hotspot stations" and show a Pleistocene age. In contrast, the background core rather refers to CO2 reduction as pathway of methane formation but this core is not at a "hotspot" location.

Line 253- 259 when methane is formed by acetate fermentation the whole methyl group is used to produce methane, which means that at least 3 of the 4 hydrogens are formerly fixed in organic matter and not in water.

Line 263 high concentration in frozen sediments just show that methane is available.

Line 310 14C values show old carbon, not clear that it is old methane

Line 331 Additional to oxidation, the isotopic signature of methane in seawater is influenced by mixing and dilution. The combination of all these processes will modify the isotopic signature. Assumptions about potential sources for thermogenic methane can neither be confirmed nor neglected. This assumption is not possible by using just this dataset.

Conclusions
Based on the discrepancies between the data and the biased interpretations the conclusions are not on a scientifically based fundament.

Line 385 this assumption is just one possibility
Line 391 this conclusion is not related to the data as this paper gives no data about gas bubbles
Line 405-410 the data shown in this paper shows neither the emission of large amounts of methane nor the role of submarine thawing permafrost for methane release to the atmosphere

Table 1 where in the text is this table discussed/mentioned?
When discussed this table should include the fractionation factor ε for this data set

---

## Short Comment (SC1) · 24 Oct 2016

Notes on Sapart et al. *The origin of methane in the East Siberian Arctic Shelf unraveled with triple isotope analysis*

The paper makes a very important point about the limitations of the isotope data to resolve sources. It should eventually be published but major revisions are required.

It is really difficult to extract the point of indistinguishable but biogenic sources. The paper looks like it has been reworked for different journals with additions and subtractions making for a confusing mix of irrelevant obfuscation and discussion. The Supplement could be removed though Figs S1 and S3 contribute to the narrative and could be added to the main text.

It should be noted by the editor that I have active projects with some of the co-authors, one of whom is a current graduate student in my laboratory.

*Some general comments*:
More information about the individual cores themselves would be very useful. What was the overlying water depth? How far from shore were they? How long ago is it estimated that they were flooded? It is very confusing throughout the paper with what samples are from frozen sediments and what are not.

The paper needs to be rewritten in a consistent and organized way. There are a number of irrelevant references. The ms is littered with irrelevant and unbalanced (in the sense of number) references while a range of recent references is not considered. The use of irrelevant and multiple in-house references are not useful.

The use of single and double quotation marks is inconsistent, baffling and distracting. Quotes should only be used for direct attribution.

I offer specific comments below to improve the presentation. Hope it helps.

*Some particular comments*:

Abstract does not express the point of the ms. The first paragraph seems out of place.

l. 35: It is unclear as to what "Large scale CH4 super saturation" refers. And it should be noted that "super saturation" is in reference to atmospheric values. I note this because it could be confused because the paper is about sediments (where bubbles are formed only after saturation with a pure CH4 atmosphere).

l. 49: This is likely to be a problem for its use as a background metric given that the contaminated core is the core to which the others are compared.

ll. 51-53: This probably should be the third sentence of the first paragraph and, at the same time, this is not a conclusion this is a motivation for the research. It's more like "might be true".

l. 67: Are all the processes mentioned not releasing subsea Pleistocene carbon?

ll.67-69: Did not know that thermokarst formation (talik deepening maybe) and active layer deepening affects *subsea* emissions. The active layer is a seasonal feature. Are you saying that there is a seasonal freeze-thaw cycle in subsea sediments?

l. 78: change "and" to "or"

l. 86: Methane oxidation is not restricted to surface sediments especially AOM. This can occur to significant depths depending upon the OM content, porosity and the availability of CH4 and SO4.

l. 90: More correctly perhaps, "It is surmised that..." As written, it confuses assumptions with observations.

ll. 97-122. It seems that there is a lot of irrelevant speculation here. Is the purpose of this ms to test if there are reservoirs of CH4 in the ESAS that are at risk of thawing? It seems that the observations of relatively high CH4 concentrations are a reason for looking into the isotopes to see if different sources can be defined.

l. 98. It might help if a calculation based on temperature and pressure can be presented for the depth where hydrates can be expected to form on the shelf and slope.

l. 102: Recent papers by Stranne and Archer might be considered here..

l. 108: The reasons for the recent warming are unclear (probably Atlantic water intrusion) and are they relevant to this discussion? The issue is the rate of deepening of permafrost thaw which is an ongoing process since the area was originally flooded rather than hydrate decomposition. Temperature data from the boreholes would be useful. Is citation of a 30 year old paper sufficiently recent? Also there are more recent assessments of hydrates in the Arctic Ocean, eg Ruppell 2014 and references therein.

l. 109: I do not understand. It is first claimed that it is extensive and now you're saying it is largely degraded?

l.117: Good to note this. It should also be noted in the abstract.

l. 120: I thought this paper was about origins of sedimentary CH4. Though they might provide a rationale for examining the isotopes, these air/sea flux references are irrelevant. No connection is made between the signatures of the CH4 dissolved in porewater and water with atmospheric values.

ll. 120-121: "high concentration" of what? High concentration of plumes? I do not remember seeing any bubble, porewater or water column concentration data specifically in the plumes in these papers. It's good enough to say that coring was done in a region with a high frequency of observed plumes. This implies that the coring was done directly in a sediment supplying a plume.

l. 122:  Could be true.  Cannot tell if sediment source refers to diffusive flux across the sediment/water boundary or from CH4-rich bubble plumes that originate deeper in the sediments traversing the water column.  This implies a diffusion-limited source from the surface sediments and is highly inconsistent with the vigorous mixing throughout the water column depth posited in the papers referenced at the beginning of this sentence.

l. 145:  "straight after" How soon after drilling? How long is "immediately"?

ll.153-155:  I am confused with the description.  All these different detectors and columns were used on one  g.c. while analyzing every sample?  What was the precision?  These different detectors will have very different response curves.
What standards were used?
Probably generalize here and provide careful DETAILS in the supplement.

l. 162: What does this add?
As it's presented it is a calculated estimate based on the total bulk density which would be sufficient. This might be important if you were trying to model variations in diffusion coefficients x tortuosity of the sediments or even try to identify regions that might be more amenable to advective  flux but this is the only place it is mentioned.

l. 173: Largest? In what sense? Concentration? Volume? Why not just give the mass of C required for each analysis and/0r what cut off you used.

l. 184: There seems to be a logic error here, how do you "expand" something into a smaller volume?

l. 187:  Not separating the Results and Discussion makes the information hard to find and the explanations very much harder to follow.

l. 191: So you're comparing four cores taken off of Tiksi with water samples that are taken 100's of km away in different water column depths and different marine environments with no physical oceanographic data to demonstrate that these environments are connected.  A word or two more here would help understand the rationalization.

l. 198: It is not clear as to why is this a background site? Especially given how contaminated the core is. So it is a "non-ebullition" site, or a "non-bubbling" site.  It is certainly not background for $^{14}$C.

l. 200: "IID-13, IIID-13 and VD-13 cores were thawed down to 19, 17 and 12m, respectively." It is not clear how the thaw front is defined (ice-bound permafrost?). Also, the thaw depth of core IIID-13 is not displayed in Figure S1.

L 206:  There is no section 3.1.

l. 206: Why "deep" here and not "surface" in the subsequent section. Use quotes only for direct attribution.

l. 216: Why 4 references here and none elsewhere for this list. And are not these specific refs more about terrestrial OM transport rather than sedimentary microbial biogeochemistry?

l. 225: "salinity measurements" Seawater in all the sediment cores below the permafrost boundary? How can they be halfway frozen? Salinity data for all cores should be shown to support this claim would be helpful.

l. 228: Why aren't the sulfate (and other chemical) data for any but the contaminated core presented? Also some indication of the frozen depth in the other cores should be presented in Fig. 2.

l. 231 and throughout: *in situ* is simply italicized – not hyphenated nor placed in quotes.

l. 232: Suppressed? It is possible given the potential presence of SO4 (though no data are given) but no evidence is presented for active inhibition which is what is implied. There is actually not so much OM. And given the references noted above (but not here), they would imply that the available OM will be heavily degraded terrestrial material and so it is not surprising that acetoclastic methanogenesis could be substrate limited – though SO4 inhibition is more likely. The presence of SO4 and AOM will also have implications for the stable isotope signatures.

l.241: Which sampling location? And how does the CH4 migrate? It could be argued from the very light del-D values and the age of the CH4 C that the CH4 is produced in place if it is hydrogenotrophic soon after thaw in the presence of freshwater and then does not migrate very fast at all. Migration pathways are an important part of the deep-production hypothesis, and a short discussion could be extracted from the SI and incorporated here in the main text.

l. 247: Another overlooked reference, Koch et al. 2008 reports distributions of methanogenic communities in subsea permafrost that might support the hypothesis of *in situ* production in partially thawed cores. This indicates that the CH4 might not have to be from migration through ice.

l. 250: What is the del-D of the frozen porewaters?

l. 254: The very light 13C values could be due to AOM recycling which could drive the 13C signatures much lighter. This is the most common explanation for very light marine CH4. E.g. Geprägs et al. 2016 has a nice explanatory figure. CO2 reduction of substrate from recent OM only gets us down to -80 or so.

l. 261: What high concentrations? These values look like permafrost values everywhere (actually a bit low) going back to Kvenvolden.

There is no need to invoke a very highly and more improbable migration through ice. And the low del-D as well are consistent with *in situ* production that hasn't moved much.

By the way, it is very difficult to see where the frozen sediment values are in Fig 2 which is relevant.

l. 266: Just curious but where was the sea level then? I have seen values of 55 m or so lower 15000 years ago.  Or is this material transported in?

l. 269: A reference to Fig 2 would fit here nicely.

l. 275: I think you are right.  It's the only explanation.  You pushed some surface contamination down core.  I think this is a serious problem with calling this core "background"  There is very little that you can compare with this.

ll. 295-297: The Overduin paper reports similar concentrations in the thawed portion of their core. However, as this last sentence is written, it misrepresents what the Overduin paper is saying that the CH4 is removed by oxidation with sulfate at the surface sediment. This is deceptive, because nearly 100% of the CH4 loss in the Overduin et al core happens at the thaw front where  SO4 intrusion is keeping up with the thaw boundary.

l. 300: The figure shows reflectors that could be ice not necessarily free gas.  It's ok (and better) to express the ambiguity.  Anyway, Figs. S1 and S3 should be incorporated into the main text.

l. 301: relatively impermeable? is that like relatively dead?  Better to use "relatively less permeable"

l. 302: I don't understand this.  Are you saying this is why there are specific plume sites? Is there evidence of free gas pressures or changes in horizontal advective/diffusive mechanisms driving the gas loss?  I can imagine it but would the fine grained pelite lithologies allow this?

l. 306: Possible but not sure it is likely.  Why are the surface seds laminated? or demonstrate distinct lithologies (Fig S1 )?  Wouldn't that (especially the fine-grained) be disrupted by vigorous advective flux.

l.312: There are no sediment age data shown - only CH4 data.

l. 328: The reasons for comparing water samples and trying to link them to cores taken 100's of km away is not articulated well.

l. 336:  What is meant by "deep Earth layers"?

l. 338:  again with the distracting quotes.  Who or what are you citing here?

And the pycnocline, and the well-known low rates of methanogenesis within it has been observed for decades,however it  is not usually found at the bottom.

l. 344: 40 cm!!  Everywhere on the ESAS?  This is simply wrong.  Should be removed
There are so many things wrong with this statement.  It might be true locally for very short periods (spring runoff?) but it cannot be true over the entire ESAS.  You can find videos on line and satellite data as well if you need demonstration.
Even the Amazon with some of the highest TSS loads in the world, the light penetrates to close to a meter.
It would also imply an extremely large deposition rate to the sediments – not seen.
It would also imply a huge role for Fe cycling in OM degradation – not seen.

L 351: A concentration vs isotope plot could be very helpful in supporting claims about oxidation.

ll. 352-353:  The isotopic values (as well as they can be seen in Fig2) in the cores are not really shifted that much.
In the under ice water column samples there does not appear to be a gradient between the deep samples and the near-surface samples, i.e. it's hard to see a "substantial" oxidation signal.
One could even argue that the very light values in the sediments is more of a signal of AOM due to C recycling (e.g. Geprägs et al. 2016).

l. 358: No sense in having both Figs 3 and 4.  Both are too busy anyway and could do with some simplification.

l. 362: Did I miss a plot of concentration vs signature?
I do not understand this argument for a number of reasons. It seems the assumption is that the same processes are acting at similar rates on the water, frozen sediments and thawed sediment samples.  We know that's not the case (AOM for instance).  If one looks at the water samples alone, they seem to follow a nice oxidation trend.

l. 387:  This is an assumption and perhaps maybe likely but No information is given on gas hydrate or gas distribution in the cored areas.

ll. 388-389:  This is an odd statement because migration of the gas is not "shown. "
Rather, an almost plausible interpretation of the data along those lines could be made.

ll. 393-394: I agree but you cite a number of papers where such "quantitative" estimates are made.  So again, please check the relevance of your references and trim those that are not needed.

ll. 404-405:  This is not consistent with the statement made in line 394.  And a prediction of large amounts of CH4 from thawing does not follow from any of the data presented in this paper.

Table 1: Not really used in the paper.  Why are values in the table expressed as fractionation factors rather than delta ratios as used everywhere in the text?

Fig.2: Is very busy and the depth scales are confusing.  I understand the challenge of trying to convey so much information on a given figure but it should be clear.
I especially miss noting the frozen depths, temperatures and chemistry (especially SO4). And the 14C of the OM.

Figs 3 and 4: It is difficult to resolve the diamond and square shapes.  Why do you even have the square shapes when I cannot find them discussed in the text.
Probably do not need both figures. Maybe replace Fig. 4 with a concentration vs isotope plot.

Fig. 5:  Why is this figure shown alone and not in comparison with the other cores.  Also it is difficult to relate the specific scale to the specific line.

Supplementary Information:

This entire first section is not useful.  Probably best if the core descriptions were removed and perhaps moved to the main text.
For example, none of the lithologies described in Arenson and Sego are related to those described in Fig. S1.

I am confused by the black lines alongside two of the cores.  Only two of the four cores had evidence of freezing? How close together were the cores?  They seem close but I can't really tell from the figure.
Cryostructures are not the same thing as frozen nor do they represent ice-bound permafrost.  Polygonal ground structures and cryoturbation are cryostructures and they are not frozen.  Relic structures can persist. That's why we know certain areas have been frozen before.  In fact, if they are persisting it is indicative that there is not a lot of advective turbation - i.e. bubbles or a lot of water - flowing through

It also seems there is a basic confusion about the salinity of the frozen interstitial fluids. If the permafrost was formed subaerially then it is likely that the ice will have very low salinities.  This is certainly indicated by the del-D values of the deeper CH4.

The relevance of the Biggar et al, study is not clear.  That study was about sands and gravels with very low moisture contents in essentially polar desert.  It has NO relevance to subsea permafrost.  it is about non-aqueous phase liquid migration.

In the SI there is discussion about higher HCs yet no data on higher hydrocarbons in the paper. (though C2+C3 could go a long way to resolving and testing the assumptions made in this ms).

You could move Figs S1 and S3 to the main body of the ms.  Those Figs are already discussed there and make up part of your narrative.

Fig. S2 can be removed. We know the core is contaminated and that is more or less ok.
No need to make up confusing stories about why. It does not matter.

Prof. Patrick Crill
Stockholm University

---

## Short Comment (SC2) · 3 Nov 2016

Comment on Sapart et al. 2016 "The origin of methane in the East Siberian Arctic Shelf unraveled with triple isotope analysis"

Brett F. Thornton, Department of Geological Sciences, Stockholm University

*For disclosure, I have active research projects with two of the coauthors, and have worked on past projects with some of the coauthors as well. However, I had nothing to do with the research behind or the drafting of the present Sapart et al. manuscript; the first time I saw it was when it appeared in Biogeosciences Discussions.*

$CH_4$ emissions from the East Siberian Arctic Shelf have been the subject of intense interest since reports of very high atmospheric $CH_4$ in the area, and later, reports that such enhancements were being driven by $CH_4$-containing bubble plumes from the seafloor. This manuscript presents a new dataset of $CH_4$ concentrations and $CH_4$ isotopologue studies in the sediment beneath 3 nearshore areas of the Laptev Sea, along with similar studies of $CH_4$ in the water column in these areas. Additional water column measurements are also provided for a far offshore site in the central Laptev Sea, near the top of the continental slope.

So, I was very interested to read this manuscript, and I strongly feel it should be published because of its unique dataset of $CH_4$ and $CH_4$ isotopologues in this region. However, there are some issues with the manuscript that should be cleared up. The other reviewers raise many important points, and I generally agree with them. The main problem with the manuscript is that it loses sight of the main results, in my opinion. The main observations are unique and should be published! See especially my comment about lines 404-407, where the reader finally is told the biggest result. The first section of the supplement needs rewritten or removed (I would vote for removed, because it's not especially critical to the arguments in the manuscript.)

Below I bring up specfic questions:

Line 45 "primary substrate glacial water" – I am not sure what "glacial water" means in this context; normally glacial water comes from glaciers... I think the authors mean water that has been frozen in to the subsea permafrost since formation, but I'm not sure. See also line 255.

Line 51-53—I don't see anything in this manuscript that says that the sediment $CH_4$ studied in this manuscript rapidly migrates through the water column. Bubbles were not trapped and analyzed. The last sentence of the abstract should be removed.

Line 66-70: "The four key mechanisms controlling the release of Pleistocene carbon from thawing subsea permafrost are gas hydrate degradation, thermokarst development, the deepening of the permafrost active layer and coastal erosion (e.g. Shakhova et al., 2005, 2009, 2010a,b, O'Connor et al.,2010, James et al., 2016)."

Several things wrong with this statement, it is talking about subsea permafrost but gives examples that only apply to land! (1) I have never heard of active layer deepening in subsea permafrost. This implies an annual freeze-thaw cycle, as happens to permafrost regions on land, not at sea. (2) Similarly, I'm not sure what undersea thermokarst is—thermokarst landscapes form due to seasonal cycling. Are you saying there are annual freeze-thaw cycles in subsea permafrost? (3) Coastal erosion does not, by definition, release carbon from subsea permafrost—it releases carbon from the eroding coast line. The entire sentence should be removed or rewritten.

Line 100-102: " Below this gas hydrate stability zone, CH4 occurs as free gas and can be advected towards the surface through faults in the sediment."
Why would this free gas not be incorporated into hydrates as it passes through the stability zone? Or are you suggesting that $CH_4$ released BELOW the gas hydrate stability zone migrates upwards so fast through the sediment that it is never trapped as hydrates?

Line 120: " vigorous bubbling" is undefined with no sense of scale.

Line 184: "expanded into a smaller flask for storage". Impossible to expand something into a smaller volume.

Line 191: The reader has no clue as to the logic behind the four core identifiers: " ID-11, IID-13, IIID-13, VD-13". To the reader, these are just random numbers and letters, and they are similar enough to be confusing. It would be far less confusing if they were simply designated background, 1,2,3 (or 1,2,3,4) in this manuscript. OR, if the core names themeselves are somehow significant, or correspond to information in other papers, that should be explained.

Line 199-201: The cores are shown in supplement Figure S1 (I think this figure should be in the main text). But in Figure S1, the white and black bars beside the cores show frozen/unfrozen. But here, it says that " IID-13, IIID-13 and VD-13" are partly frozen, yet IIID-13 is shown as completely thawed in figure S1.

Line 215: Another good reason to move figure S1 to the main text!

Line 255: Here, old water frozen in the permafrost is called "meteoritic"—so again, why is it called "glacial water" on line 45?

Line 282: " strong evidences that CH4 from old reservoirs (Pleistocene age or older) is being released there."  -- or that the $CH_4$ is being formed from old carbon being released from reservoirs.

Line 296-297: This seems like a slight misunderstanding of the Overduin et al paper; in that core, almost all the loss of $CH_4$ occurred at the thaw front, not near the sediment surface / seawater interface. Compare Figure 4 in the Overduin et al paper with your Figure 2. I see no sharp cutoff in $CH_4$ values at the thaw front in your paper as Overduin et al report. (However, the use of a log plot in Figure 2 makes it somewhat hard to see.) Also, label the thaw front in Figure 2 for each core.

Line 340-345: " In the ESAS, the pycnocline is very shallow and a very low primary production is expected because of darkness and ice cover in the winter and because of the little available sunlight in the summer due to the high solar zenith angles and the very turbid waters (light penetrates only down to 40cm)"

**The statements about light penetration depth are NOT TRUE. As written, this is about the entire ESAS.** I suppose that close to shore, turbid water can occur (and can be seen from space), but farther from shore, water is not so turbid (again, can be seen from space). 40 cm light penetration depth is extremely shallow. Yes, water surfaces are more reflective at shallow light incidence angles, but there is a lot of sunlight in the summer in the Arctic! Photos have been published showing blue waters around islands in the study area in the summer. I don't know if there is in situ production of $CH_4$ in the water column or not, but I am 100% certain that light penetrates far deeper than 40 cm in waters of the ESAS.

Also, the authors have previously claimed (in this journal!) that the ESAS waters have high productivity! dx.doi.org/10.5194/bg-8-1745-2011

Line 385-387: " This gas is formed continuously from old substrates at depth and/or has been stored as gas hydrate and/or gas pockets in or below the subsea permafrost."

This sounds like you are100% ruling out biogenic $CH_4$ production in the near-seafloor sediment, where such production might happen utilizing recently deposited carbon sources mobilized from terrestrial and coastal erosion sources? Interesting that you rule that out. Really?

Line 393-394: "No quantitative estimate of this CH4 source is to date possible,"
Some of the papers you cite give quantitative estimates. So do some papers you don't cite. Maybe you mean something else here?

Line 404-407: " Our results show that thawing subsea permafrost emits large amounts of CH4 that is depleted in heavy isotopes and that such emissions cannot be easily distinguished from Arctic wetland emissions when looking only at stable isotope data. "

I believe this is the most important result of the study, and this potential caveat about isotopic studies in this region should also be mentioned in the abstract, and earlier in the text.

Other items:

I do not understand why we are presented Figure 5, which shows sulfate, $C_{org}$, chloride, and Si for only the background core. Why not for all 4 of the cores discussed here? Or, better still, because the dataset presented here is unique and will be of interest to many, I **strongly encourage the authors to make the all data shown in Figure 2 available with the manuscript**, perhaps as a supplement.

I find the title a bit curious. Saying "unraveled" suggests that the mystery has been solved; I would say it has not been (but that is okay!!). To me, there appears to be vast areas of the ESAS which have not been sampled yet for sediment $CH_4$. Hence, the title seems--premature. Unless the authors mean to imply that our understanding is being unraveled?

Supplementary Material

Line 5-9: Doesn't make much sense as written. Perhaps the authors mean something like: " Although thawing is the most obvious factor affecting the permeability of permafrost to gases, there are other factors to consider, which we discuss below"

Line 16: "the content of unfrozen water"—should be "the fraction of unfrozen water"

Line 22-23: "as it has been demonstrated"—should be "as has been demonstrated"

Line 27-31: Doesn't make sense as written. Groundwater and porewater are not the same but are apparently used interchangably here.

Suggest something like: "The salinity of this cryogenic porewater usually ranges between 10 and 300psu. Freezing-point depression is also due to the dissolved-solids content of this cryogenic porewater (Gilichinsky et al., 2007). The high salinity and solids content is due to inclusion of brines from the freezing of marine sediments."

Lne 32-34: Doesn't make sense as written. Suggest something like: "These water layers are usually connected to each other, building up a multi-level transport system which allows gases and geofluids to migrate through subsea permafrost and potentially be released to the water column, possibly via taliks."

Lines 35-38: redundant.

Line 35: Biggar et al study is not relevant here. It's about (as the title gives away) spilled fuels migrating downwards.

Line 40, Section 1.2.

This seems to be about terrestrial permafrost; but this section is headed "Factors affecting gas transport in subsea permafrost".

Line 44: "alterations of compression" doesn't make much sense. Maybe something like " they affect frozen soils and sediments by alternately compressing and stretching them during freeze-thaw cycles."

Line 53: Section 1.3.

This entire section is messy and difficult to read. It is also about processes happening far below the study zone of this manuscript. In my opinion, it can be removed without any loss to the manuscript.

Line 140-141: The problem with explaining the 14C-hot samples is that they are hottest at depth, right? Why would anthropogenic contamination not be greater at the top of the sediment, instead of under 30 m of sediment? That is a mystery. Seems like some comment should be made about this (at least to acknowledge the mystery.)

Line 155 "was abnormal" should be "were abnormal".

Figure S1: Should be part of main text. Label which core is the background core.

---

## Author Comment (AC1) · 9 Dec 2016

Please find the final Response to the review as supplement

Please also note the supplement to this comment:
http://www.biogeosciences-discuss.net/bg-2016-367/bg-2016-367-AC1-supplement.pdf

---

## Author Comment (AC2) · 9 Dec 2016

**1) FIRST REVIEWER:**

The manuscript (ms) by Sapart et al. presents valuable information on the origin of methane in the East Siberian Arctic Shelf (ESAS), long been discussed based on assumptions and speculations. I like that this data set represents results of triple-isotope analysis. To my knowledge, this is first presentation of such a kind collected in the marine Arctic. This fact itself makes this ms novel. Besides, the data are from the potentially globally important region, because a predominant fraction of submarine permafrost is there; the ESAS was suggested to incorporate much of organic carbon and, probably, hydrocarbon stocks and hydrates in the sediments beneath permafrost. It was also shown to be warming due to natural warming as well as global warming. I believe this new data would be of great interest to scientists working in different disciplines and areas of the Arctic: to geologists, bio-geochemists, oceanographers, atmospheric scientists, climatologists and climate scientists.

Authors report methane concentrations and triple isotope data analyzed on gas extracted from sediment (four long cores down to ←·· 53m depth) and water sampled over the ESAS from 2007 to 2013. Samples were taken from different areas of the ESAS, which represent different state of subsea permafrost (frozen to thaw) extending from the coastal zone to the outer shelf. The manuscript is clearly written. The approach is valid and the applied techniques are appropriate – I have no questions regarding triple isotope measurements accomplished by the authors in the best European laboratories. I can't completely agree with the authors that that the predominant presence of methane is not of thermogenic/natural gas origin, but resultant from microbial methane formation using as primary substrate glacial water and old carbon preserved in the subsea permafrost or below, but I can share the logic of the authors of this ms, which is mostly resultant from limitations possessed by currently available methods of triple-isotope data interpretation.

*1)AUTHORS: We thank reviewer 1 for his/her comments.We agree that our triple isotope dataset does not allow us to totally exclude the presence of thermogenic methane in the ESAS sediment. But we state that at the sediment sampling locations, the methane present in the sediment porewater is clearly of biogenic origin and no thermogenic signatures have been observed there.*

It is interesting that most of methane in the ESAS sediments is of Pleistocene age or older – in my understanding, this should mean that submarine permafrost is somehow permeable for gases. I am not satisfied with explanations presented by the authors regarding the contribution of super-modern radiocarbon in methane – anthropogenic sources would have been explaining this fact plausibly if it has been methane from the water column, not from the sediment cores. I understand that it must be very complicated to come up with more realistic explanations, but the authors should be working further on it.

*2)AUTHORS: As we stated in the main text (lines 267-274) and in more detail in the SI (section 3), 14C levels >200 pmC do not occur in nature and have never been introduced by atmospheric nuclear bomb tests, the most common source of super-modern 14C levels from 100-200 pmC in the environment from the 1950s until today. It should be noted that the unit pmC refers to the 14C/12C isotopic ratio of a sample related to the ratio of a standard value of the "modern" level. Due to the usage of 14C/12C isotopic ratios, any geochemical transformation of carbon will keep its pmC value. It should be further noted that for the sediment samples, the*

*largest enrichment in 14C is observed at 30m depth in the seabed thus sea water cannot be the cause of this enrichment. We believe that the most likely hypothesis to explain this highly enriched 14C values is that nuclear wastes have been deposited somewhere in the permafrost (likely inland) and that leakages from this area are contaminating the groundwater aquifer and thus lateral underground transport may transfer organic matter or gases highly enriched in 14C to the subsea permafrost.*

*However, we agree that it is very complicated to corroborate our explanations by independent analyses. We have had, together with all co-authors and other specialists in nuclear physics, substantial discussions on alternative approaches. It came out that only data based on other radionuclides could help us to identify more precisely the origin of this enrichment, but such measurements were not possible to perform with the samples we had in hands at the time we found out about this issue. Therefore, a special field campaign to cover a larger area (terrestrial and marine) and aiming to extract much larger volume of sediments would have been required to obtain such data, but we did not have either the funding nor the authorization for such a deployment.*

**Specifically**:

- Why did not authors consider results published by Franke and Cramer (2005) and Bussmann et al., (2013), who presented clearly thermogenic signature of methane sampled in the same area?

*3)AUTHORS: These publications as well as the one of Preuss et al.,2013 will be discussed in the section 3.5.*

-Why did not authors collected gas from bubbles included in the sea ice (reported by Shakhova et al., 2010) to measure triple-isotope signature of methane? Would not it give a kind of integral isotope signature of methane potentially approaching the atmosphere?

*4)AUTHORS: We have tried many times to sample gas from the bubbles (directly in the water and in extracting the gas from the ice on the field), but that is highly complicated (much more than on lakes), because of the harsh conditions on the sea ice in the winter and the lack of stability while working on the ice. We have a system to measure methane isotopes on ice samples in our laboratory (see Sapart et al., AMT, 2011) that has been adapted and used for sea ice samples, but the transport of ice samples from Russia to Europe has never worked.*

-Why did not authors consider release of super-modern methane found further off-shelf (lines 327) by groundwater transported from the land (lines 327-329)?

*5)AUTHORS: This is indeed our main hypothesis (nuclear contamination transported in the groundwater through the thawing permafrost layers to the subsea permafrost and sea water) (see comment 2 for more details). This will be rewritten in a clearer way in the discussion.*

- How would the authors explain enrichment of d13C by <50 per mil if residence time of shelf water is only few months?

*6)AUTHORS: We do not understand to what the 50pmil enrichment refers to, because we do not observe such an enrichment at any of our water sampling locations. Nevertheless, an enrichment in heavy isotopes could be explained:*

- *by removal and/or diffusion processes if it is accompanied by a decrease in concentrations, however diffusive fractionation would be generally very small.*

- *by the adding of methane from a  more enriched reservoir if the concentration increase.*
- *by a mixture of both processes.*

Many other questions could be raised, but I realize that this manuscript is based on multi-year work in the harsh Arctic environment. It is clear to me, that one paper, even incorporating that extensive data set, cannot answer all scientific questions regarding the complex, and previously unstudied, Arctic marine methane cycle. I appreciate that the authors have been accumulating data for a long period trying to cover as much aspects of this novel topic as possible. I also understand limitations possessed by current state of isotope biogeochemistry, which make it difficult to interpret isotope data collected in actual environmental conditions where methane of different origin, age, from different sources could be contributing differently in different areas – it differs so much from all idealized conceptions used to interpret the data. I suggest that these questions would be addressed in further work on this topic and the current ms would be taken as a baseline, relative to which results of further investigations in this area could be evaluated. At this point, I recommend that this paper is published as is.

**2) SECOND REVIEWER:**

**Overall:**

The paper: "The origin of methane in the East Siberian Shelf unraveled by triple isotopic analysis" reports isotopic data from methane in sediments and seawater. This unique data set comprises methane   13C, 14C and   D values with concentrations enabling an unusual insight into isotopic shifts between sediments and sea water but also between different sediment cores in this polar region. The data clearly reveal the predominance of biogenic methane. Longterm speculations about thermogenic/natural gas methane sources could not be confirmed. Beside this basic news the most surprising outcome is the fact that methane in sediments of the ESAS shelf is much more depleted in $^{13}$C and D than expected when considering the two main pathways of methane formation, i.e. acetate fermentation and CO2 reduction. Unfortunately the interpretation of this unique data set is biased by just focusing to prove the existence of methane diffusing from larger depths by thawing permafrost to the sediment surface and further through the water column up to the atmosphere. There is no doubt about the existence of carbon sources from Pleistocene age. Old particulate organic matter derived from permafrost soils, ice complexes and coastal erosion and transported by the Lena represents an important contribution to Siberian shelf sediments. Hence the option of a more recent methane formation with old C (terrestrial C?) should be at least also discussed. Pleistocene aged water is argued to be involved in methane formation by CO2 reduction. While I am able to follow that argument of upward diffusing of Pleistocene aged methane as one possibility, the data are ambiguous and should be discussed in a broader context.

*7)AUTHORS: We would like to thank Reviewer 2 for his/her comments. At reading the review, we have realized that the quality of our discussion has suffered from the numerous edits and shortening the paper went through, therefore section 3 will be restructured and a thorough discussion on the different sources of carbon in the ESAS will be added.*

Especially as both the $\delta$13C and the $\delta$D values are clearly out of the range of bacterial methane formation by both pathways, i.e. CO2 reduction and acetate formation. The samples shown here have either $\delta$D values or $\delta$13C values untypical for the proposed pathways. The shift in deltaD values from hotspot cores to the background core is conspicuous. I would appreciate a discussion of this circumstance. Furthermore, the hotspot sediment cores show deltaD values, which are in a "normal" range for acetate formation, while just the $\delta$13C values vary in a broader range. However, this pathway is unable to prove inclusions of old glacial water in methane. In contrast, the background core rather refers to CO2 reduction as pathway of methane formation but this core is not localized at a "hotspot" position and also not Pleistocene aged. Especially in the context of the hydrogen sources in methane, further non-competitive pathways are worthwhile to be discussed. Summarized, it remains questionable if the isotopic signature of methane (especially the $\delta$D values as evidence for the CO2 reduction pathway) in the sediments is really 1:1 related to the former pathway of methane formation or if there are additional fractionation effects which modify the final signature.

*8)AUTHORS: The main factor causing the difference between the d13C of the background and hotspot cores must be the origin and the state of the carbon substrate. The background core has a clear CO2 reduction d13C signature, but some methane formed by acetate fermentation could add to the CO2-reduced methane pool which would involve the range observed for the hotspot cores. We state that for the background core, methane cannot be formed in situ, because sea water has percolated all the way down the core. This means that:*
  1) *sulfate is present all along the core and will inhibit acetate fermentation at the depth where the core was drilled.*
  2) *seawater would be the substrate for methane in the case it is produced by carbonate reduction insitu. But that is not the case as shown by the dD data of the background core.*
*For the hotspot cores, because of harsh conditions in the field, we could not obtain very good salinity profiles, so it remains unclear how deep exactly the seawater has mixed with meltwater from buried meteoric ice. There, we might have a mixture of acetate fermentation and carbonate reduction using old carbon as substrate as shown by the 14CH4 results. However, if we had a mixture of CO2 reduced methane (using as substrate present mean ocean water) and methane formed by acetate fermentation, our sediment data point (the diamonds) should be between the dotted lines on Fig.3. This is not the case, because dD is about -130pmil more depleted in heavy isotopes. The only way for methane to have such depleted deuterium signatures is to use a substrate (water in the case of CO2 reduction) very depleted in deuterium. To our knowledge, meltwater from buried meteoric ice is then the only possible substrate sufficiently depleted in deuterium to explain the dD data we obtained for both background and hotspot cores.*

*We have done investigations on the role of diffusive transport on the fractionation, and the few data we have found show that the fractionation involved was small in with the signal we measured for both stable isotope signatures (Fuex, 1979, Prinzhofer and Pernaton, 1997,Chanton et al. 2005)*

Concerning the sea water data, the main result is to see the decoupling between sediments and water above. Without any information on the bathymetric regime any

interpretation remains speculative.
*9)AUTHORS: The table below with detailed information on each drilling site will be added to the revised manuscript.*

| Borehole/shelf area of sampling | Latitude | Longitude | Water depth, m | Water temperature surface/ bottom | Water salinity surface/ bottom | Time since inundation | Distance from the shore, km |
|---|---|---|---|---|---|---|---|
| **ID-11** | 71,6926 | 130,3669 | 12.5 | -0.52/-0.92 | 10.8/19.4 | ~8÷7 kyr BP | 18.6 km from Muo-stakh Isl and 39.8 km from Bykovsky Peninsula |
| **IID-13** | 71,6288 | 129,8534 | 2.5 | -0.41/-0.61 | 12.8/13.2 | ~7÷6 kyr BP | 2.5 km |
| **IIID-13** | 71,6219 | 129,8517 | 4.1 | -0.40/-0.61 | 12.8/13.2 | ~7÷6 kyr BP | 2.5 km |
| **VD-13** | 71,7449 | 129,4048 | 3.6 | -1.01/-1.03 | 22.6/22.6 | Lagoon, ~7÷6 kyr BP (Romanovsky et al., 1999) | Lagoon adjacent to the coast |
| **Lena Delta** Str. 42* | 72,977 | 130,141 | 12.5 | 3.74/1.42 | 16.3/22.3 | ~7÷6 kyr BP | - |
| 47* | 72,427 | 130,151 | 8.5 | 4.42/2.53 | 0.5/4.6 | | |
| 70** | 72.687 | 130.640 | 20 | 2.40/2.83 | 19.0/21.5 | | |
| 71** | 72.683 | 130.267 | 12 | 1.02/2.47 | 9.0/19.1 | | |
| **Dmitry Laptev** Str. 31* | 72,882 | 140,620 | 15.5 | 3.38/3.76 | 18.7/20.3 | ~8÷7 kyr BP | - |
| 36* | 73,118 | 139,517 | 13.5 | 3.30/3.44 | 19.8/24.0 | | |
| 79** | 73.030 | 139.393 | 15.0 | 1.92/3.04 | 19.8/23.3 | | |
| 80** | 73.169 | 139.574 | 15 | 1.67/2.42 | 19.3/26.7 | | |
| 81** | 73.316 | 139.768 | 9 | 1.85/1.79 | 18.6/24.8 | | |
| 82** | 73.297 | 140.085 | 11 | 1.6/1.46 | 24.17/26.4 | | |
| 83** | 73.104 | 140.352 | 14 | 1.75/2.19 | 20.3/26.5 | | |
| 84** | 72.930 | 140.619 | 12 | 1.77/1.77 | 21.1/21.1 | | |
| 85** | 72.886 | 140.643 | 24 | 1.69/1.77 | 20.8/20.9 | | |
| Buor-Khaya Gulf | Drilling sites | | | | | ~8÷7 kyr BP | - |
| **Shelf edge** Sts. 9* | 76,897 | 127,8047 | 62 | 3.54/-1.33 | 28.0/33.4 | ~15÷13 kyr BP | - |
| 13* | 76,894 | 127,8032 | 63 | 3.47/-1.40 | 29/1/33.3 | | |
| 28** | 77.000 | 125.985 | 91 | 3.87/-0.82 | 28.8/34.4 | | |
| 32** | 77.029 | 127.289 | 74 | 3.00/-1.21 | 21.9/34.2 | | |
| 36** | 76.398 | 125.051 | 62 | 1.26/-1.17 | 19.0/34.1 | | |
| 37** | 76.648 | 125.048 | 70 | 1.89/-1.19 | 17.3/34.2 | | |

Although I completely agree that this data set raises much more new questions than it is able to answer, a careful interpretation of this data set would improve the quality of the discussion.

Finally, it would be worthwhile to revise the conclusions and those should be based on the data rather than on speculations which remain to be proven at this stage. I recommend major revisions.

**Details:**

**Abstract**
Line 30-37 is an introduction only remotely related to the data discussed here
Line 49-53 this assumption is not proved by the data shown here

*10)AUTHORS: The abstract has been rewritten as follows:*
       "The Arctic Ocean, especially the East Siberian Arctic Shelf (ESAS) has been proposed as a significant source of atmospheric methane that might play an increasingly important role in the future. However, the processes of formation, removal and transport associated with such emissions are to date strongly debated.
       In this study, $CH_4$ concentrations, stable isotope ratios and radiocarbons were analyzed on gas extracted from sediment and water sampled over the shallow ESAS from 2007 to 2013. We found high concentrations (up to 500µM) of $CH_4$ in the pore water of the partially thawed subsea permafrost of this region. For all sediment cores, both hydrogen and carbon $CH_4$ isotope data reveal the predominant presence of $CH_4$ that is not of thermogenic/natural gas origin as it has long been thought, but resultant from microbial $CH_4$ formation using as primary substrate meltwater from buried meteoric ice and old organic matter preserved in the subsea permafrost or below. Radiocarbon data demonstrate that the $CH_4$ present in the ESAS sediment is of Pleistocene age or older, but a small contribution of highly $^{14}C$-enriched $CH_4$, from unknown origin, prohibits precise age determination for one sediment core and in the water column. Our data suggest that at locations where bubble plumes have been observed, $CH_4$ can escape anaerobic oxidation in the surface sediment and be released directly to the water column."

**Introduction**
In general a short introduction in using delta13C and deltaD values is needed.
Processes which might modify the signature additionally to pathways of formation are completely missing and should be included.
*11)AUTHORS: We have added the paragraph below:* "Each type of $CH_4$ formation/removal pathway produces $CH_4$ with a characteristic isotopic signature ($\delta^{13}C$ and $\delta D$) depending on the isotopic composition of the substrate and the isotopic fractionation associated with the respective chemical reaction involved. Microbes take up more easily lighter isotopologues hence $CH_4$ produced by methanogenesis has a lighter isotopic signature than its substrates but when it is consumed, its remaining reservoir will become isotopically more enriched in heavy isotopes (e.g. Whiticar 1999, Conrad, 2005). Similarly, diffusive transport can cause isotopic discrimination because lighter isotopologues diffuse faster than heavier ones. However, this fractionation is considered to be relatively small (<5pmil: Fuex, 1979, <20pmil: Prinzhofer and Pernaton, 1997 and 3pmil: Chanton et al. 2005) in comparison with the isotopic fractionation associated with methanogenesis (7-95pmil for $\delta^{13}C$ and 260-430pmil for $\delta D$) and with $CH_4$ oxidation (2-39pmil for $\delta^{13}C$ and

66-350pmil for δD) (Whiticar, 1999, Holler et al., 2009)"

Line 65- concerning the remobilization of carbon from the Pleistocene: terrestrial carbon transported by the Lena River into the shelf sediments should also be mentioned, see: Winterfeld et. al., 2015 Biogeoscience).

*12)AUTHORS:This sentence has been modified as follows:* "The four suggested key mechanisms controlling the release of Pleistocene carbon to the ESAS are gas hydrate degradation, the deepening of the permafrost level, coastal erosion and riverine (e.g. Shakhova et al., 2005, 2009, 2010a,b, O'Connor et al., 2010, Winterfeld et al., 2015, James et al., 2016)."

Line 78 Further pathways for methanogenesis should be at least mentioned (methylotrophic with non-competitive substrates- see also Whiticar et al, 1999).
*13)AUTHORS:This has been added.*

Line 91-96 The "Whiticar scheme" considering the relationship between the isotopic composition and pathway of methane formation has been developed for sediments. Using this scheme for sea water data should include at least a discussion about further fractionation effects in order to avoid over-interpretation of data. For example: Methane diffusion through sediments might induce fractionation effects just described by Prinzhover and Pernaton 1997.
*14)AUTHORS:This has been added as discussed in comment 11).*

Further, there is no proof that the isotopic signature of methane dissolved in sea water (outside a plume) can be used in a simple way for source identification of methane formation in sediments or from gas fields.
*15)AUTHORS: We did not do such a claim in our ms and we do not believe that sea water can be used in a simple way for source identification, because as written in the ms (l.372-380), in shallow areas, the gas bubbles from the sediment do not have time to equilibrate totally with the gas dissolved in the seawater hence, our water measurement represent a mixture of what is coming from the sediment and from potential other sources (riverine and aerobic formation in the surface water). As explained in the ms (l. 336-350), aerobic surface production is unlikely, however, we will add a paragraph about the potential input of riverine methane in discussing our data together with the data of Preuss et al., 2013, Bussmann et al., 2013 and Franke and Cramer, 2005.*

Line 97-122 gives a detailed description about potential methane source at the ESAS, however, that paragraph about potential gas hydrates and gas bubbles in that region does not introduce the topic of the paper and is not a helpful tool to understand the isotopic data shown here. Further, this paragraph is misleading as the data shown in the paper are not in that range to push forward the knowledge about the role of gas hydrates and gas bubbles and are not needed to introduce the paper.
Line 115 this citation is wrong in that context. Measured concentrations should be related to the atmospheric equilibrium concentration in ESAS.I recommend adding supplement information into the introduction (physical factors) as this contribution is essential for understanding and interpretation of this data set.
*16)AUTHORS: L 97-122 are removed and replaced by a corrected section (physical factors) of the SI.*

**Methods**
Line 173 the term "largest samples" is misleading and should be improved.
*17)AUTHORS: largest samples refers to sample with at least 20micrograms of methane. That will be added.*

**Results and Discussion**

For my knowledge these is the first data set of methane isotopic signatures in frozen sediments hence potential freezing effects on isotopic fractionation should at least be taken into account.
Line 213-215 how differences in lithology influence differences in the isotopic signature should be discussed in the main text.
*18)AUTHORS: To our knowledge, no data exist on the role of the lithology on the physical fractionation of gases therefore a discussion on this matter would be speculative. The meaning of this sentence (l.213-215) is that according to the type of lithology (hence of sediment) different types (or amount) of substrates could be available at the different location. This sentence will be rephrased.*

Line 219-222 most of the isotopic signatures of the deep sediment cores are not included in the range of CO2 reduction or acetate formation. Potential reasons should be discussed.
Line 223and Figs 3 and 4: yes, methane is unusually depleted in 13C and D. The samples shown here have either deltaD values or delta$^{13}$C values untypical for the proposed pathways. More in detail: for acetate formation the 13C values are untypical while the deltaD are in a "normal"range. However, this pathway is unable to prove the inclusion of old glacial water. But these cores are from the "hotspot stations" and show a Pleistocene age. In contrast, the background core rather refers to CO2 reduction as pathway of methane formation but this core is not at a "hotspot" location.
*19)AUTHORS: This section will be reworked. See comment 8).*

Line 253- 259 when methane is formed by acetate fermentation the whole methyl group is used to produce methane, which means that at least 3 of the 4 hydrogens are formerly fixed in organic matter and not in water.
*20)AUTHORS: We state that methane is likely formed by CO2 reduction using meltwater from buried meteoric ice or (and) by acetate fermentation, but we do not imply that meltwater from buried meteoric ice is used in acetate fermentation.*
Line 263 high concentration in frozen sediments just show that methane is available.
*21)AUTHORS: This sentence will be rephrased as follows: "High $CH_4$ concentrations are also observed in frozen sediment showing the presence of $CH_4$ in the permafrost layers."*
Line 310 14C values show old carbon, not clear that it is old methane.
*22)AUTHORS: The 14C values presented here are 14C-CH4 data. It shows that methane is formed on old substrate and that is what we mean by old methane. This will added to the revised version.*

Line 331 Additional to oxidation, the isotopic signature of methane in seawater is influenced by mixing and dilution. The combination of all these processes will modify the isotopic signature. Assumptions about potential sources for thermogenic methane can neither be confirmed nor neglected. This assumption is not possible by using just

this dataset.

*23)AUTHORS: We agree with this comment and we have been careful not drawing clear conclusions on the presence of thermogenic methane. In l.335, we write: Thermogenic emissions are possible, …..but we have not measured it at our sediment sites"*

**Conclusions**

Based on the discrepancies between the data and the biased interpretations the conclusions are not on a scientifically based fundament.

Line 385 this assumption is just one possibility

Line 391 this conclusion is not related to the data as this paper gives no data about gas bubbles.

Line 405-410 the data shown in this paper shows neither the emission of large amounts of methane nor the role of submarine thawing permafrost for methane release to the atmosphere.

*24)AUTHORS: the conclusion has been rewritten in focusing on the key results according to the suggestions of some of the reviewers. The conclusion is now written as follows:* "Our triple isotope dataset of $CH_4$ from the sediment and water of the shallow ESAS reveals the presence of $CH_4$ of biogenic origin formed on old carbon with unexpectedly low carbon ($\delta^{13}C$ as low as -108‰) and hydrogen ($\delta D$ as low as -350‰) signatures down to about 50m below the seabed in the thawed permafrost. Stable isotope data demonstrate that at locations where a thick marine clay layer is present, this $CH_4$ is partially oxidized before reaching the seawater. However at locations where ebullition was observed from the seabed, no oxidation was identified in the stable isotope surface sediment profile. In that case and considering the very shallow water column (<10m) in this area, this biogenic gas will likely reach the atmosphere when sea ice is absent. Our results show that thawing subsea permafrost of the ESAS emits $CH_4$ with an isotopic signature that cannot be easily distinguished from Arctic wetland emissions when looking only at stable isotope data. Hence, it remains very difficult to quantify both processes independently using atmospheric $CH_4$ stable isotope measurements and isotopic mass balance models."

Table 1 where in the text is this table discussed/mentioned?

*25)AUTHORS: As required by P. Crill in his short comment, Fig. 4 has been replaced by a concentration/isotope plot, therefore this table is not used anymore and is therefore removed.*

When discussed this table should include the fractionation factor epsilon for this data set.

**3) SHORT COMMENT BY PATRICK CRILL:**

The paper makes a very important point about the limitations of the isotope data to resolve sources. It should eventually be published but major revisions are required.

It is really difficult to extract the point of indistinguishable but biogenic sources. The paper looks like it has been reworked for different journals with additions

and subtractions making for a confusing mix of irrelevant obfuscation and discussion. The Supplement could be removed though Figs S1 and S3 contribute to the narrative and could be added to the main text.

It should be noted by the editor that I have active projects with some of the co-authors, one of whom is a current graduate student in my laboratory.

**Some general comments**:

More information about the individual cores themselves would be very useful. What was the overlying water depth? How far from shore were they? How long ago is it estimated that they were flooded? It is very confusing throughout the paper with what samples are from frozen sediments and what are not.

*26)AUTHORS: A table (see comment 9) has been added with all the information required for each core and that will be discussed in the section 2.1.*

The paper needs to be rewritten in a consistent and organized way. There are a number of irrelevant references. The ms is littered with irrelevant and unbalanced (in the sense of number) references while a range of recent references is not considered. The use of irrelevant and multiple in-house references are not useful.

The use of single and double quotation marks is inconsistent, baffling and distracting.

Quotes should only be used for direct attribution.

*27)AUTHORS: The abstract (see comment 10), part of the introduction, section 3.2 and 3.5 as well as the conclusion (see comment 24) have been rewritten and restructured according to the suggestions of the referees.*

I offer specific comments below to improve the presentation. Hope it helps.

**Some particular comments:**

Abstract does not express the point of the ms. The first paragraph seems out of place.

l. 35: It is unclear as to what "Large scale CH4 super saturation" refers. And it should be noted that "super saturation" is in reference to atmospheric values. I note this because it could be confused because the paper is about sediments (where bubbles are formed only after saturation with a pure CH4 atmosphere).

l. 49: This is likely to be a problem for its use as a background metric given that the contaminated core is the core to which the others are compared.

ll. 51-53: This probably should be the third sentence of the first paragraph and, at the same time, this is not a conclusion this is a motivation for the research. It's more like "might be true".

*28)AUTHORS: The abstract has been rewritten (see comment 10) and the four comments above have been taken into account.*

l. 67: Are all the processes mentioned not releasing subsea Pleistocene carbon?

*29)AUTHORS: Yes as it is written in the ms, the four processes mentioned are allowing the remobilization of subsea Pleistocene carbon.*

ll.67-69: Did not know that thermokarst formation (talik deepening maybe) and active layer deepening affects subsea emissions. The active layer is a seasonal feature. Are you saying that there is a seasonal freeze-thaw cycle in subsea sediments?

*30)AUTHORS: This sentence has been corrected and rewritten (see comment 12).*

l. 78: change "and" to "or"
*31)AUTHORS: this has been corrected.*
l. 86: Methane oxidation is not restricted to surface sediments especially AOM. This can occur to significant depths depending upon the OM content, porosity and the availability of CH4 and SO4.
*32)AUTHORS: the word "surface" has been removed.*

l. 90: More correctly perhaps, "It is surmised that..." As written, it confuses assumptions with observations.
*33)AUTHORS: The fact that each type of biological or chemical reactions to produce methane (or other trace gases) is associated to characteristic isotopic fractionation factors is not an assumption, but something that is well established. Therefore, we do not believe that it is appropriate to write "It is surmised that…" in that context.*

ll. 97-122. It seems that there is a lot of irrelevant speculation here. Is the purpose of this ms to test if there are reservoirs of CH4 in the ESAS that are at risk of thawing? It seems that the observations of relatively high CH4 concentrations are a reason for looking into the isotopes to see if different sources can be defined.
l. 98. It might help if a calculation based on temperature and pressure can be presented for the depth where hydrates can be expected to form on the shelf and slope.
l. 102: Recent papers by Stranne and Archer might be considered here..
l. 108: The reasons for the recent warming are unclear (probably Atlantic water intrusion) and are they relevant to this discussion? The issue is the rate of deepening of permafrost thaw which is an ongoing process since the area was originally flooded rather than hydrate decomposition. Temperature data from the boreholes would be useful. Is citation of a 30 year old paper sufficiently recent? Also there are more recent assessments of hydrates in the Arctic Ocean, eg Ruppell 2014 and references therein.
l. 109: I do not understand. It is first claimed that it is extensive and now you're saying it is largely degraded?
l.117: Good to note this. It should also be noted in the abstract.
l. 120: I thought this paper was about origins of sedimentary CH4. Though they might provide a rationale for examining the isotopes, these air/sea flux references are irrelevant. No connection is made between the signatures of the CH4 dissolved in porewater and water with atmospheric values.
ll. 120-121: "high concentration" of what? High concentration of plumes? I do not remember seeing any bubble, porewater or water column concentration data specifically in the plumes in these papers. It's good enough to say that coring was done in a region with a high frequency of observed plumes. This implies that the coring was done directly in a sediment supplying a plume.
l. 122: Could be true. Cannot tell if sediment source refers to diffusive flux across the sediment/water boundary or from CH4-rich bubble plumes that originate deeper in the sediments traversing the water column. This implies a diffusion-limited source from the surface sediments and is highly inconsistent with the

vigorous mixing throughout the water column depth posited in the papers referenced at the beginning of this sentence.

*34)AUTHORS: Considering the comments above and the suggestions of the second reviewers, this paragraph (l.97-122) has been removed.*

l. 145: "straight after" How soon after drilling? How long is "immediately"?

*35)AUTHORS: Straight after and immediately mean here a few minutes. This has been corrected in the revised ms.*

ll.153-155: I am confused with the description. All these different detectors and columns were used on one g.c. while analyzing every sample? What was the precision? These different detectors will have very different response curves. What standards were used?

*36)AUTHORS: The GC method we have used is very commonly used, therefore we did not add details in the ms about it. Here is a more detailed paragraph which could be added in the Supplementary Information if requested: " The GC used to measure CH₄ concentrations has two 10-Port Gas Sampling Valves, 2 meter MoleSeive 13X column, 30 m capillary column and 6 channel PeakSimple usb data system and was equipped with a flame ionization detector (FID), which was used for concentrations of CH₄ <200 ppmv, and a thermal Conductivity Detector (TCD), which was used for concentrations of CH₄ >200 ppm. The GC oven was operated isothermally (40˚C) and the maximum detector temperature was held at ≈250˚C. The carrier gas used was helium. Daily calibration was performed with a certified 1.96 ppmv and 99.999 ppmv CH₄ gas standard with the air (Air Liquide, USA). The standard deviation of duplicate analyses (three to five replicates) was <2%. Reproducibility was ~1% based on multiple standard injections during daily calibrations. The standard deviation of duplicate analyses (3-5 replicates) was less than 2%. GC precision had standard error of only 1%. The concentration of dissolved CH₄ in the water and sediment samples was calculated with the Bunsen solubility coefficient for CH4 [Wiesenburg and Guinasso, 1979] for the appropriate equilibration temperature, pressure and the volume of headspace and water/sediment in the bottles. "*

Probably generalize here and provide careful DETAILS in the supplement.

l. 162: What does this add?

As it's presented it is a calculated estimate based on the total bulk density which would be sufficient. This might be important if you were trying to model variations in diffusion coefficients x tortuosity of the sediments or even try to identify regions that might be more amenable to advective flux but this is the only place it is mentioned.

*37)AUTHORS: This sentence has been removed from the ms.*

l. 173: Largest? In what sense? Concentration? Volume? Why not just give the mass of C required for each analysis and/0r what cut off you used.

*38)AUTHORS: The sample containing more than 20micrograms of CH₄. This has been added as follows: "Radiocarbon analyses could be performed only on the largest samples (containing more than 20micrograms of CH₄)"*

l. 184: There seems to be a logic error here, how do you "expand" something into a smaller volume?

*39)AUTHORS: Here we meant that part of the sample was transferred "by expansion" from the burette to a smaller volume. The word "expanded has been replaced by transferred" in the revised ms.*

l. 187: Not separating the Results and Discussion makes the information hard to find and the explanations very much harder to follow.

*40)AUTHORS: We have done several attempts to write the results and the discussion separately, but that had obliged us to make a lot of repetitions. Therefore, we have done the choice to write one section Results and discussions and we believe that it is the most appropriate manner to present our data. However, we have worked on the structure of section 3 to discuss each point in a more structured and thorough way.*

l. 191: So you're comparing four cores taken off of Tiksi with water samples that are taken 100's of km away in different water column depths and different marine environments with no physical oceanographic data to demonstrate that these environments are connected. A word or two more here would help understand the rationalization.

*41)AUTHORS: These different entities belong to the same shelf system, which begins from the near-shore (where we drilled) and extends to the shelf edge (where we sampled water), but please note that most of our water samples were taken close to the coast where we also sampled surface sediment (see Fig.2) This entire shelf was exposed above the sea level during the last glacial period, froze to the depth of few hundred meters and was believed to been keeping its integrity until recent times. So the most quintessential, fundamental part of this marine ecosystem, which builds up the unity of this environment, is subsea permafrost, which is underlying seafloor - a unique component of this marine environment. Investigating different parts of the shelf is crucially important because current state of subsea permafrost depends on duration of inundation and this duration varies from <1000 years in the near-shore zone to >10 000 years in the outer shelf and the shelf edge. The ideal approach would be to sample water and recover long sediment cores in each of observed settings. The problem is that drilling at greater water depths requires incomparably greater funding than what scientists usually can obtain from scientific funding agencies. Moreover, the very harsh conditions during the winter (when the deeper cores were drilled) did not allow us to reach the shelf edge. Therefore, we present the existing data, which are the results of many years of fieldwork in this region as a first attempt to better understand the origin of the methane in the coastal ESAS.*

l. 198: It is not clear as to why is this a background site? Especially given how contaminated the core is. So it is a "non-ebullition" site, or a "non-bubbling" site. It is certainly not background for 14C.

*42)AUTHORS: We named the site where no ebullition was observed a background site, before the $^{14}C$ enrichment was observed and we have kept this denomination. However, to help the reader to grasp the main difference, we changed this denomination to "non-ebullition site" in the revised version.*

l. 200: "IID-13, IIID-13 and VD-13 cores were thawed down to 19, 17 and 12m, respectively." It is not clear how the thaw front is defined (ice-bound permafrost?). Also, the thaw depth of core IIID-13 is not displayed in Figure S1.

*43)AUTHORS: A more detailed scheme for each core (as shown below as example) is added to replace Fig. S1 and this scheme are discussed in section 3.1 of the revised manuscript.*

Water depth 3.6 m

| Sediment | Cryostratigraphy and cryogenic condition | Description |
|---|---|---|
| | cryotic | Black mud. |
| | thawed | Silty loam, dark-grey. |
| | | Coarse-grained sand, black. |
| | | Loamy sand, dark-grey. |
| | | Silty loam, dark-grey. Deeper 7.8 m - frozen. Cryostructure lenticular with ice lenses thickness up to 1 cm. |
| | | Fine-grained sand, dark-grey. Cryostructure is structureless, deeper 13.15 m sand is cryotic. |
| | cryotic | Coarse-grained sand, dark-grey. Upper part is cryotic, deeper 19.7 m - frozen, cryostructure is structureless. |
| | | Fine-grained sand, dark-grey and greenish. Cryostructure is structureless. |
| | | Coarse-grained sand with rare pebbles up to 5 cm, dark-grey. Cryostructure is structureless. |

L 206: There is no section 3.1.

*44)AUTHORS: this has been corrected in the revised ms.*

l. 206: Why "deep" here and not "surface" in the subsequent section. Use quotes only for direct attribution.

*45)AUTHORS: This has been corrected in the revised ms.*

l. 216: Why 4 references here and none elsewhere for this list. And are not these specific refs more about terrestrial OM transport rather than sedimentary microbial biogeochemistry?

*46)AUTHORS: We wrote that differences in isotope composition of our samples could be dependent on a variety of factors, among which one factor (the heterogeneity in substrate availability), for which we provided four citations. It appears that this first point has been more investigated in this study area than the three others and that is why more references are added there.*

l. 225: "salinity measurements" Seawater in all the sediment cores below the permafrost boundary? How can they be halfway frozen? Salinity data for all cores should be shown to support this claim would be helpful.

*47)AUTHORS: We have corrected this sentence as follows: "However, salinity measurements along the ID-2011 core indicate the presence of interstitial seawater all the way down the core." And this section is rewritten as discussed in comment 8. The conditions on the field and the funding availability for the 2013 campaign (when the hotspot cores were drilled) were very different to the 2011 campaign (when the "background core" was drilled). In 2013, the weather conditions did not allow for much safe time on the ice, which prevented us to perform good quality salinity measurements and sampling for other biogeochemical species when the core was extracted. However, considering that no other study has yet been published on this area showing more than one isotopic signature of methane in one sediment core, we believe that our dataset brings substantial information to allow for a better understanding of the origin of methane in different sectors of the ESAS.*

l. 228: Why aren't the sulfate (and other chemical) data for any but the contaminated core presented? Also some indication of the frozen depth in the other cores should be presented in Fig. 2.

*48)AUTHORS: Please see comment 43 and 47.*

l. 231 and throughout: in situ is simply italicized – not hyphenated nor placed in quotes.

*49)AUTHORS: This has been corrected in the revised ms.*

l. 232: Suppressed? It is possible given the potential presence of SO4 (though no data are given) but no evidence is presented for active inhibition which is what is implied. There is actually not so much OM. And given the references noted above (but not here), they would imply that the available OM will be heavily degraded terrestrial material and so it is not surprising that acetoclastic methanogenesis could be substrate limited – though SO4 inhibition is more likely. The presence of SO4 and AOM will also have implications for the stable isotope signatures.

*50)AUTHORS: As discussed in comment 8, this part is rewritten and the role of acetate fermentation and the difference in substrate availability is discussed further.*

l.241: Which sampling location? And how does the CH4 migrate? It could be argued from the very light del-D values and the age of the CH4 C that the CH4 is produced in place if it is hydrogenotrophic soon after thaw in the presence of freshwater and then does not migrate very fast at all. Migration pathways are an important part of the deepproduction hypothesis, and a short discussion could be extracted from the SI and incorporated here in the main text.

*51)AUTHORS: In the revised manuscript, this sections has been rewritten in discussing more in detailed the different hypotheses (transport and formation) that could involve a depletion in dD.*

l. 247: Another overlooked reference, Koch et al. 2008 reports distributions of methanogenic communities in subsea permafrost that might support the hypothesis of in situ production in partially thawed cores. This indicates that the CH4 might not have to be from migration through ice.

*52)AUTHORS: Koch et al, 2008 have showed the presence of methanogenic communities in the subsea permafrost and they indeed claim that permafrost thaw ignites methanogenesis in sediments. This supports our interpretation, but contradicts the claims of Overduin et al., that methanogenesis does not occur in thawed sediments but rather oxidation does. We will add a paragraph of discussion about in the revised ms.*

l. 250: What is the del-D of the frozen porewaters?

*53)AUTHORS: We do not have such measurements, however we believe that it is appropriate to assume that the dD data of permafrost meltwater measured by e.g. Brosius et al., 2012 in North Siberia is of the same range as the dD of the meltwater of the ESAS subsea permafrost, because the latitude difference between the study locations is small and these regions have been exposed to relatively similar conditions during the last glacial period.*

l. 254: The very light 13C values could be due to AOM recycling which could drive the 13C signatures much lighter. This is the most common explanation for very light marine CH4. E.g. Geprägs et al. 2016 has a nice explanatory figure. CO2 reduction of substrate from recent OM only gets us down to -80 or so.

*54)AUTHORS: Thank you for the suggestion, this reference as well as a short discussion about it has been added to section 3.1. of the revised ms.*

l. 261: What high concentrations? These values look like permafrost values everywhere (actually a bit low) going back to Kvenvolden.

There is no need to invoke a very highly and more improbable migration through ice. And the low del-D as well are consistent with in situ production that hasn't moved much. By the way, it is very difficult to see where the frozen sediment values are in Fig 2 which is relevant.

*55)AUTHORS: To our knowledge, Kvenvolden et al has never reported methane concentrations in the frozen sediment, so we do not believe that our data can be compare to this work. dD-CH4 data cannot be interpreted alone and at looking at both stable isotope signatures, especially for the core ID-2011 where we have a clear CO2 reduction d13C signature and where sea water is present along the whole core, the dD data show that it not possible that methane was formed there using sea water as substrate. Our data cannot give information on how deep it was formed, but it was not formed at the depth where the core was drilled.  This section 3.2. has been totally rewritten and this issue is discussed further in the revised ms.*

l. 266: Just curious but where was the sea level then? I have seen values of 55 m or so lower 15000 years ago. Or is this material transported in?

*56)AUTHORS: To our knowledge, such information are not precisely known, but our sediment sampling locations where in the state of terrestrial permafrost during the Pleistocene period.*

l. 269: A reference to Fig 2 would fit here nicely.

*57)AUTHORS:This has been added to the revised ms.*

l. 275: I think you are right. It's the only explanation. You pushed some surface contamination down core. I think this is a serious problem with calling this core "background" There is very little that you can compare with this.

*58)AUTHORS: We are convinced that this $^{14}$C enrichment is not coming from the water or sediment surface, because the largest enrichment is at about 30m depth in the sediment (Fig.2d) and at this site, there is a thick marine clay layer that would not facilitate downward migration. This situation also clearly excludes the possibility of a contamination of the samples during coring, which would have caused a trend of decreasing pmC values with increasing sediment depths, whereas we observed exactly the opposite trend (Fig. 2d). See additional explanations on the definition of the unit pmC and its relevance for the interpretation of the location of the contamination origin in comment 2.*

portion of their core. However, as this last sentence is written, it misrepresents what the Overduin paper is saying that the CH4 is removed by oxidation with sulfate at the surface sediment. This is deceptive, because nearly 100% of the CH4 loss in the Overduin et al core happens at the thaw front where SO4 intrusion is keeping up with the thaw boundary.

*59)AUTHORS: This has been corrected in the revised manuscript.*

l. 300: The figure shows reflectors that could be ice not necessarily free gas. It's ok (and better) to express the ambiguity. Anyway, Figs. S1 and S3 should be incorporated into the main text.

*60)AUTHORS: The difference between gas and ice is indeed difficult to identify with such acoustic techniques. However, in our case, the entire sediment core (ID-2011) recovered from the borehole was thawed to the depth of 52 m as seen from Fig. S1(the left-most core). Therefore, the acoustic anomaly observed in the seismic image could not be ice.*

l. 301: relatively impermeable? is that like relatively dead? Better to use "relatively less permeable"

*61)AUTHORS: This has been corrected in the revised ms.*

l. 302: I don't understand this. Are you saying this is why there are specific plume sites? Is there evidence of free gas pressures or changes in horizontal advective/diffusive mechanisms driving the gas loss? I can imagine it but would the fine grained pelite lithologies allow this?

*62)AUTHORS: Here we state that gas is accumulating under the thick marine clay layer as shown by our acoustic and methane concentration data and we assume that part of it must migrate horizontally and could potentially be released from the sediment to the water at locations where the marine clay layer is thinner or absent. This is rephrased in the revised manuscript.*

l. 306: Possible but not sure it is likely. Why are the surface seds laminated? or demonstrate distinct lithologies (Fig S1 )? Wouldn't that (especially the fine grained) be disrupted by vigorous advective flux.

*63)AUTHORS: We write that ebullition may disturb the surface layers (that would explain our sediment surface stable isotope data, but the processes involved there are not yet understood. Vigorous advection may also play a role and that will be added in the revised version.*

l.312: There are no sediment age data shown - only CH4 data.

*64)AUTHORS: This has been corrected in the revised ms.*

l. 328: The reasons for comparing water samples and trying to link them to cores taken 100's of km away is not articulated well.

*65)AUTHORS: Section 3.5 has been totally rewritten and restructured in order to discuss more thoroughly the different types of methane sources possible in the ESAS coastal water and in adding more references as discussed in comment 15.*

l. 336: What is meant by "deep Earth layers"?

*66)AUTHORS: Here we mean the Earth's crust. That has been corrected in the revised ms.*

l. 338: again with the distracting quotes. Who or what are you citing here?
And the pycnocline, and the well-known low rates of methanogenesis within it has been observed for decades,however it is not usually found at the bottom.

*67)AUTHORS: The quotation mark has been removed in the revised manuscript*

l. 344: 40 cm!! Everywhere on the ESAS? This is simply wrong. Should be removed There are so many things wrong with this statement. It might be true locally for very short periods (spring runoff?) but it cannot be true over the entire ESAS. You can find videos on line and satellite data as well if you need demonstration. Even the Amazon with some of the highest TSS loads in the world, the light penetrates to close to a meter.
It would also imply an extremely large deposition rate to the sediments – not seen. It would also imply a huge role for Fe cycling in OM degradation – not seen.

*68)AUTHORS: Here, we discuss about the shallow coastal area where most of our water samples were sampled. This has been rephrased as follows: "In the near-shore ESAS (depth <30 m, about 75% of the total area), where most of our water samples were collected, the pycnocline is very shallow and a very low primary production is expected…." (see also comment 99).*

L 351: A concentration vs isotope plot could be very helpful in supporting claims about oxidation.

*69)AUTHORS: A concentration vs isotope plot (see below) has been added to replace Fig. 4 and is discussed in the revised version of section 3.5 methane in the water.*

[Figure]

ll. 352-353: The isotopic values (as well as they can be seen in Fig2) in the cores are not really shifted that much.

In the under ice water column samples there does not appear to be a gradient between the deep samples and the near-surface samples, i.e. it's hard to see a "substantial" oxidation signal.

One could even argue that the very light values in the sediments is more of a signal of AOM due to C recycling (e.g. Geprägs et al. 2016).

*70)AUTHORS: This sentence is removed from the revised ms and the potential role of AOM is discussed further in the revised 3.5 section.*

l. 358: No sense in having both Figs 3 and 4. Both are too busy anyway and could do with some simplification.

*71)AUTHORS:Fig. 4 is replaced by a concentration vs isotope plot (see comment 69).*

l. 362: Did I miss a plot of concentration vs signature?

I do not understand this argument for a number of reasons. It seems the assumption is that the same processes are acting at similar rates on the water, frozen sediments and thawed sediment samples. We know that's not the case (AOM for instance). If one looks at the water samples alone, they seem to follow a nice oxidation trend.

*72)AUTHORS: The concentration vs isotope plot (see comment 69) shows that most of the water samples do not show a nice oxidation trend. This is discussed in the revised version of section 3.5.*

l. 387: This is an assumption and perhaps maybe likely but No information is given on gas hydrate or gas distribution in the cored areas.

ll. 388-389: This is an odd statement because migration of the gas is not "shown. " Rather, an almost plausible interpretation of the data along those lines could be made.

ll. 393-394: I agree but you cite a number of papers where such "quantitative" estimates are made. So again, please check the relevance of your references and trim those that are not needed.

ll. 404-405: This is not consistent with the statement made in line 394. And a prediction of large amounts of CH4 from thawing does not follow from any of the data presented in this paper.

*73)AUTHORS: The conclusion has been rewritten (see comment 24) in order to focus on the main message of the paper and to answer to the 4 comments above and to the suggestions of other reviewers.*

Table 1: Not really used in the paper. Why are values in the table expressed as fractionation factors rather than delta ratios as used everywhere in the text?

*74)AUTHORS: Table 1 was used to give the range of fractionation factor for Fig.4. This figure has been removed from the revised ms, hence table 1 is removed as well.*

Fig.2: Is very busy and the depth scales are confusing. I understand the challenge of trying to convey so much information on a given figure but it should be clear.
I especially miss noting the frozen depths, temperatures and chemistry (especially SO4). And the 14C of the OM.

*75)AUTHORS: We have done many attempts to produce a clear figure to show all our data at once, and this figure came out as the best possible figure we could obtain. As noted in comment 47, we do not have the all set of biogeochemical and physical data for each core, but we have shown everything we had available and that was relevant for the interpretation of our data in Fig.2 and in Fig.5. We do not believe that combining these two figures would make any improvement on the readability of the figure.*

Figs 3 and 4: It is difficult to resolve the diamond and square shapes. Why do you even have the square shapes when I cannot find them discussed in the text.

*76)AUTHORS: The square shapes are a mistake and are removed in the revised ms.*

Probably do not need both figures. Maybe replace Fig. 4 with a concentration vs isotope plot.

*77)AUTHORS: That has been done, see comment 69.*

Fig. 5: Why is this figure shown alone and not in comparison with the other cores. Also it is difficult to relate the specific scale to the specific line.

*78)AUTHORS: see comment 47.*

Supplementary Information:
This entire first section is not useful. Probably best if the core descriptions were removed and perhaps moved to the main text.

*79)AUTHORS: This has been done in the revised ms.*

For example, none of the lithologies described in Arenson and Sego are related to those described in Fig. S1. I am confused by the black lines alongside two of the cores. Only two of the four cores had evidence of freezing? How close together were the cores? They seem close but I can't really tell from the figure. Cryostructures are not the same thing as frozen nor do they represent ice-bound permafrost. Polygonal ground structures and cryoturbation are cryostructures and they are not frozen. Relic structures can persist. That's why we know certain areas have been
frozen before. In fact, if they are persisting it is indicative that there is not a lot of advective turbation - i.e. bubbles or a lot of water - flowing through. It also seems there is a basic confusion about the salinity of the frozen interstitial fluids.

*80)AUTHORS:Fig.S1 will be replaced by better schemes (see comment 43) to describe more thoroughly the structure of each core.*

If the permafrost was formed subaerially then it is likely that the ice will have very low salinities. This is certainly indicated by the del-D values of the deeper CH4.

*81)AUTHORS: Indeed, that could be expected, but this would be the case only if seawater has not reached yet the thawed permafrost layers.*

The relevance of the Biggar et al, study is not clear. That study was about sands and gravels with very low moisture contents in essentially polar desert. It has NO relevance to subsea permafrost. it is about non-aqueous phase liquid migration.

*82)AUTHORS: This section has been removed from the SI.*

In the SI there is discussion about higher HCs yet no data on higher hydrocarbons in the paper. (though C2+C3 could go a long way to resolving and testing the assumptions made in this ms).

*83)AUTHORS: According to the suggestions of all reviewers, we have focused the revised discussion on our key isotopic signature results. Therefore, this paragraph will be removed from the revised SI.*

You could move Figs S1 and S3 to the main body of the ms. Those Figs are already discussed there and make up part of your narrative.

Fig. S2 can be removed. We know the core is contaminated and that is more or less ok. No need to make up confusing stories about why. It does not matter.

*84)AUTHORS: Fig.S1 is now much larger and includes much more details for each core (as requested by the reviewers), but we believe that these information are complementary but do not need to be in the main manuscript to understand the main message. Therefore we believe it is better to keep this figure with a detailed discussion on the lithology in the SI. However, Fig. S3 is added to the revised main manuscript and Fig. S2 is removed.*

**4) COMMENT BRETT THORNTON:**

For disclosure, I have active research projects with two of the coauthors, and have worked on past projects with some of the coauthors as well. However, I had nothing to do with the research behind or the drafting of the present Sapart et al. manuscript; the first time I saw it was when it appeared in Biogeosciences Discussions.

CH4 emissions from the East Siberian Arctic Shelf have been the subject of intense interest since reports of very high atmospheric CH4 in the area, and later, reports that such enhancements were being driven by CH4-containing bubble plumes from the seafloor. This manuscript presents a new dataset of CH4 concentrations and CH4 isotopologue studies in the sediment beneath 3 nearshore areas of the Laptev Sea, along with similar studies of CH4 in the water column in these areas. Additional water column measurements are also provided for a far offshore site in the central Laptev Sea, near the top of the continental slope.

So, I was very interested to read this manuscript, and I strongly feel it should be published because of its unique dataset of CH4 and CH4 isotopologues in this region. However, there are some issues with the manuscript that should be cleared up. The other reviewers raise many important points, and I generally agree with them. The main problem with the manuscript is that it loses sight of the main results, in my opinion. The main observations are unique and should be published! See especially my comment about lines 404-407, where the reader finally is told the biggest result.

*85)AUTHORS: We thank Dr Thornton for his comments and suggestions. We agree that the key message of the paper is coming too late in the paper and that has corrected in the revised version of the paper. The abstract has been rewritten (see*

*comment 10) and some part of the introduction have been replaced to focus more on what needs to be introduced to understand the main message of the paper. Then section 3 has been largely rewritten so that the main message will be clearly stated in this section already and the conclusion section has been rewritten (see comment 24).*

The first section of the supplement needs rewritten or removed (I would vote for removed, because it's not especially critical to the arguments in the manuscript.)

*86)AUTHORS: This section has been removed from the revised version.*

**Specific questions**:

Line 45 "primary substrate glacial water" – I am not sure what "glacial water" means in this context; normally glacial water comes from glaciers... I think the authors mean water that has been frozen in to the subsea permafrost since formation, but I'm not sure. See also line 255.

*87)AUTHORS: Here we mean water from buried ice (probably segregation ice) of meteoric origin since it has initially been formed by precipitation that have infiltrated the ground and then refrozen. We replace glacial water by " meltwater from buried meteoric ice" in the revised version.*

Line 51-53—I don't see anything in this manuscript that says that the sediment CH4 studied in this manuscript rapidly migrates through the water column. Bubbles were not trapped and analyzed. The last sentence of the abstract should be removed.

*88)AUTHORS: This sentence has been removed from the revised version, see comment 10.*

Line 66-70: "The four key mechanisms controlling the release of Pleistocene carbon from thawing subsea permafrost are gas hydrate degradation, thermokarst development, the deepening of the permafrost active layer and coastal erosion (e.g. Shakhova et al., 2005, 2009, 2010a,b, O'Connor et al.,2010, James et al., 2016)."

Several things wrong with this statement, it is talking about subsea permafrost but gives examples that only apply to land! (1) I have never heard of active layer deepening in subsea permafrost. This implies an annual freeze-thaw cycle, as happens to permafrost regions on land, not at sea. (2) Similarly, I'm not sure what undersea thermokarst is—thermokarst landscapes form due to seasonal cycling. Are you saying there are annual freeze-thaw cycles in subsea permafrost? (3) Coastal erosion does not, by definition, release carbon from subsea permafrost—it releases carbon from the eroding coast line. The entire sentence should be removed or rewritten.

*89)AUTHORS: This sentence has been corrected and rewritten as follows adding the Winterfeld reference as requested by other reviewers: "The four key mechanisms controlling the release of Pleistocene carbon to the ESAS are gas hydrate degradation, the deepening of the ice-bonded permafrost table, coastal erosion and riverine (e.g. Shakhova et al., 2005, 2009, 2010a,b, O'Connor et al., 2010, Wintereld et al., 2015, James et al., 2016)."*

Line 100-102: " Below this gas hydrate stability zone, CH4 occurs as free gas and can be advected towards the surface through faults in the sediment." Why would this free gas not be incorporated into hydrates as it passes through the stability zone? Or are you suggesting that CH4 released BELOW the gas hydrate stability zone migrates upwards so fast through the sediment that it is never trapped as hydrates?

*90)AUTHORS: This section has been removed, see comment 16.*

Line 120: " vigorous bubbling" is undefined with no sense of scale.

*91)AUTHORS: "vigorous bubbling" is defined in the paper cited in the same*

*sentence (Shakhova, N, et al. The East Siberian Arctic Shelf: towards further assessment of permafrost-related methane fluxes and role of sea Ice. Phil. Trans. R. Soc. A 373, 20140451 (2015) and more detailed are given there on how the bubbling was observed.*

Line 184: "expanded into a smaller flask for storage". Impossible to expand something into a smaller volume.

*92)AUTHORS: Here we meant that part of the sample was transferred "by expansion" from the burette to a smaller volume.  The word "expanded has been replaced by transferred" in the revised ms.*

Line 191: The reader has no clue as to the logic behind the four core identifiers: " ID-11, IID-13, IIID-13, VD-13". To the reader, these are just random numbers and letters, and they are similar enough to be confusing. It would be far less confusing if they were simply designated background, 1,2,3 (or 1,2,3,4) in this manuscript. OR, if the core names themeselves are somehow significant, or correspond to information in other papers, that should be explained.

*93)AUTHORS:The name of the cores depend on the year they were drilled, so 1D-2011, was the first core of 2011, and then we have second core of 2013 (IID-2013), third core of 2013 (IIID-2013) and fifth core of 2013 (VD-2013). This is how the cores are referred to therefore we believe it is appropriate to keep this denomination in this paper in order to be consistent with the work of others.*

Line 199-201: The cores are shown in supplement Figure S1 (I think this figure should be in the main text). But in Figure S1, the white and black bars beside the cores show frozen/unfrozen.

*94)AUTHORS: Fig.S1 has been replaced by more detailed schemes (see comment 43). Therefore, Fig.S1 is now much larger and includes much more details for each core (as requested by the reviewers), but we believe that these information are complementary but do not need to be in the main manuscript to understand the main message. Therefore we believe it is better to keep this figure with a detailed discussion on the lithology in the SI.*

But here, it says that " IID-13, IIID-13 and VD-13" are partly frozen, yet IIID-13 is shown as completely thawed in figure S1.

*95)AUTHORS: This has been corrected in the revised version.*

Line 215: Another good reason to move figure S1 to the main text!

*96)AUTHORS: see comment 94.*

Line 255: Here, old water frozen in the permafrost is called "meteoritic"—so again, why is it called "glacial water" on line 45?

*97)AUTHORS: Here we mean meltwater from buried meteoric ice, thus water derived from precipitation during glacial times.*

Line 282: " strong evidences that CH4 from old reservoirs (Pleistocene age or older) is being released there." -- or that the CH4 is being formed from old carbon being released from reservoirs.

*98)AUTHORS: In this case, we mean methane that is formed (or was formed) using old carbon reservoirs. Triple isotope data cannot allow to identify when exactly the methane was formed, but on which type of substrate. That will be clarified in the revised version.*

Line 296-297: This seems like a slight misunderstanding of the Overduin et al paper; in that core, almost all the loss of CH4 occurred at the thaw front, not near the sediment surface / seawater interface. Compare Figure 4 in the Overduin et al paper with your Figure 2. I see no sharp cutoff in CH4 values at the thaw front in your paper as Overduin et al report. (However, the use of a log plot in Figure 2 makes it

somewhat hard to see.) Also, label the thaw front in Figure 2 for each core.

*99)AUTHORS: This has been clarified and discussed more thoroughly in the revised manuscript.*

Line 340-345: " In the ESAS, the pycnocline is very shallow and a very low primary production is expected because of darkness and ice cover in the winter and because of the little available sunlight in the summer due to the high solar zenith angles and the very turbid waters (light penetrates only down to 40cm)".

The statements about light penetration depth are NOT TRUE. As written, this is about the entire ESAS. I suppose that close to shore, turbid water can occur (and can be seen from space), but farther from shore, water is not so turbid (again, can be seen from space). 40 cm light penetration depth is extremely shallow. Yes, water surfaces are more reflective at shallow light incidence angles, but there is a lot of sunlight in the summer in the Arctic! Photos have been published showing blue waters around islands in the study area in the summer. I don't know if there is in situ production of CH4 in the water column or not, but I am 100% certain that light penetrates far deeper than 40 cm in waters of the ESAS.

*100)AUTHORS: See comment 68). Since the depth of penetration can vary from one place to another, we remove the 40cm from the revised version. See Fig. S5 of Semiletov et al., Acidification of East Siberian Arctic Shelf water through addition of freshwater and terrestrial carbon. Nature Geosciences, 18 April 2016, doe:10.1038/NGEO2695. They show the distribution of SPM in the surface water of the ESAS that gives information on the turbidity of the surface water at some locations where our water sampling was carried out. This reference will be added to the discussion in the revised version.*

Also, the authors have previously claimed (in this journal!) that the ESAS waters have high productivity! dx.doi.org/10.5194/bg-8-1745-2011.

*101)AUTHORS: Some of the authors of this paper have cited other studies, who investigated rates of productivity in different areas of the ESAS. For example, Sorokin (1996) reported rates in the Lena River estuary that were moderately high in the Lena Delta but drop orders of magnitude shelf ward. In the Chukchi Sea, there is also a well-known spot of high productivity, where whales found their feeding habitat. But they did not claim high levels of primary productivity (PP) in the ESAS.*

Line 385-387: " This gas is formed continuously from old substrates at depth and/or has been stored as gas hydrate and/or gas pockets in or below the subsea permafrost." This sounds like you are100% ruling out biogenic CH4 production in the nearseafloor sediment, where such production might happen utilizing recently deposited carbon sources mobilized from terrestrial and coastal erosion sources? Interesting that you rule that out. Really?

*102)AUTHORS: The conclusion has been rewritten, please see comment 24.*

Line 393-394: "No quantitative estimate of this CH4 source is to date possible,"
Some of the papers you cite give quantitative estimates. So do some papers you don't cite. Maybe you mean something else here?

*103)AUTHORS: The conclusion has been rewritten, please see comment 24.*

Line 404-407: " Our results show that thawing subsea permafrost emits large amounts of CH4 that is depleted in heavy isotopes and that such emissions cannot be easily distinguished from Arctic wetland emissions when looking only at stable isotope data." I believe this is the most important result of the study, and this potential caveat about isotopic studies in this region should also be mentioned in the abstract, and

earlier in the text.

*104)AUTHORS: This has been put forward earlier in the text and in the revised conclusion, see comment 24.*

Other items:

I do not understand why we are presented Figure 5, which shows sulfate, Corg, chloride, and Si for only the background core. Why not for all 4 of the cores discussed here?

*105)AUTHORS: Please see comment 47.*

Or, better still, because the dataset presented here is unique and will be of interest to many, I strongly encourage the authors to make the all data shown in Figure 2 available with the manuscript, perhaps as a supplement.

*106)AUTHORS: We will make all the data shown in Figure 2 available when the paper will be published.*

I find the title a bit curious. Saying "unraveled" suggests that the mystery has been solved; I would say it has not been (but that is okay!!). To me, there appears to be vast areas of the ESAS which have not been sampled yet for sediment CH4. Hence, the title seems--premature. Unless the authors mean to imply that our understanding is being unraveled?

*107)AUTHORS: We do not believe that our dataset allows to solve fully the issues, but it brings the knowledge and the understanding further. Therefore, we propose the following revised title: "The origin of methane in the East Siberian Arctic Shelf investigated with triple isotope analysis".*

**Supplementary Material**

Line 5-9: Doesn't make much sense as written. Perhaps the authors mean something like: "Although thawing is the most obvious factor affecting the permeability of permafrost to gases, there are other factors to consider, which we discuss below".

Line 16: "the content of unfrozen water"—should be "the fraction of unfrozen water".

Line 22-23: "as it has been demonstrated"—should be "as has been demonstrated".

Line 27-31: Doesn't make sense as written. Groundwater and porewater are not the same but are apparently used interchangably here. Suggest something like: "The salinity of this cryogenic porewater usually ranges between 10 and 300psu. Freezing-point depression is also due to the dissolved-solids content of this cryogenic porewater (Gilichinsky et al., 2007). The high salinity and solids content is due to inclusion of brines from the freezing of marine sediments."

Lne 32-34: Doesn't make sense as written. Suggest something like: "These water layers are usually connected to each other, building up a multi-level transport system which allows gases and geofluids to migrate through subsea permafrost and potentially be released to the water column, possibly via taliks."

Lines 35-38: redundant.

Line 35: Biggar et al study is not relevant here. It's about (as the title gives away) spilled fuels migrating downwards.

Line 40, Section 1.2. This seems to be about terrestrial permafrost; but this section is headed "Factors affecting gas transport in subsea permafrost".

Line 44: "alterations of compression" doesn't make much sense. Maybe something like " they affect frozen soils and sediments by alternately compressing and stretching them during freezethaw cycles."

Line 53: Section 1.3. This entire section is messy and difficult to read. It is also about processes happening far below the study zone of this manuscript. In my opinion, it can be removed without any loss to the manuscript.

*108)AUTHORS: Section 1 of the Supplementary Material has been removed.*

Line 140-141: The problem with explaining the 14C-hot samples is that they are hottest at depth, right? Why would anthropogenic contamination not be greater at the top of the sediment, instead of under 30 m of sediment? That is a mystery. Seems like some comment should be made about this (at least to acknowledge the mystery.)

*109)AUTHORS: Please see comment 58 and comment 2.*

Line 155 "was abnormal" should be "were abnormal".

*110)AUTHORS:that has been corrected in the revised version*

Figure S1: Should be part of main text. Label which core is the background core.

*111)AUTHORS: Please see comment 84.*

---

## Author Comment (AC3) · 9 Dec 2016

Please find the Final Response to the review as supplement.

Please also note the supplement to this comment:
http://www.biogeosciences-discuss.net/bg-2016-367/bg-2016-367-AC3-supplement.pdf

---

## Referee Report (RR1)

**General comments**

It was a challenge for Sapart with co-authors to satisfy entirely all the reviewers (including online open comments), but they rewrote the manuscript extensively and remade some figures to address successfully all questions raised by reviewers in their reviews**.** The revised manuscript (ms) by Sapart et al. presents valuable information on the origin of methane in the East Siberian Arctic Shelf (ESAS), long been discussed based on assumptions and speculations. I like that this data set represents multi-year results of triple-isotope analysis of water and sediment. To my knowledge, this is first presentation of such a kind collected in the marine Arctic. I made my search and found that up to date only one paper described an application of the triple-isotope analyses addressed to the marine methane origin (Kessler, J.D. et al., 2008. A survey of methane isotope abundance (C-14, C-13, H-2) from five nearshore marine basins that reveals unusual radiocarbon levels in subsurface waters. J. Geophys. Res., Oceans113. http://dx.doi.org/10.1029/2008JC004822.). This fact itself makes this ms novel as I already mentioned in my previous review. .

**Specific comments**

Now I am satisfied with explanations presented by the authors regarding my key questions:

1) **The possible contribution from thermogenic sources.** I agree with authors that their triple-isotope dataset does not allow to totally exclude the presence of thermogenic methane in the ESAS sediment in another locations. They clearly state that "at the sediment sampling locations, the methane present in the sediment porewater is clearly of biogenic origin and no thermogenic signatures have been observed there". I think that that result combined with the literature data (e.g. Cramer and Franke, 2005; Bussmann et al., 2013) might indicate on high heterogeneity of sources in the vast ESAS region.

2) **The possible contribution of super-modern radiocarbon in methane by anthropogenic sources.** Authors demonstrated that the largest enrichment in 14C is observed at about 30m depth in the seabed suggesting that sea water cannot be the cause of this enrichment. I also agree that the most likely hypothesis to explain this highly enriched 14C values is that nuclear wastes have been deposited somewhere in the permafrost (likely inland) and that leakages from this area are contaminating the groundwater aquifer and therefore lateral underground transport may transfer organic matter highly enriched in 14C to the shelf environment including subsea permafrost. From my knowledge, I would add (no need to include in this ms) that one from sources of that enrichment could be numerous underground nuclei bombing tests performed in the northern Yakutia in the end of Soviet epoch.

And again, as I wrote in my first review: "Many other questions could be raised, but I realize that this manuscript is based on multi-year work in the harsh Arctic environment. It is clear to me, that one paper, even incorporating that extensive data set, cannot answer all scientific questions regarding the complex, and previously insufficiently studied, Arctic marine methane cycle. I appreciate that the authors have been accumulating data for a long period trying to cover as much aspects of this novel topic as possible. I also understand limitations possessed by current state of isotope biogeochemistry, which make it difficult to interpret isotope data collected in actual environmental conditions where methane of different origin, age, from different sources could be contributing differently in different areas – it differs so much from all idealized conceptions used to interpret the data". I suggest that these questions would be addressed in further work on this topic and the current ms would be taken as a baseline, relative to which results of further investigations in this area could be evaluated.

**Bottomline**. By my opinion it is very complicated to corroborate their explanations by independent analyses. Therefore, a special field campaign to cover a larger area (terrestrial and marine) and aiming to

extract much larger volume of water and sediments would have been required to obtain such data, but authors have either insufficient funding nor the authorization for such a deployment. Note, that getting a set of permissions to work in the Russian Exclusive Economic Zone (EEZ) needs a lot of long-term efforts and a good fortune. I admire authors' persistence exhibited in order to obtain such unique results in the Russian EEZ, including new data from the long sediment cores recovered from boreholes drilled by author's team.

I believe this new data would be of great interest to scientists working in different disciplines and areas of the Arctic: to geologists, biogeochemists, oceanographers, atmospheric scientists, climatologists and climate scientists. At this point, and to stimulate further development of this novel, original, and complex research , I recommend to make the results presented in this paper available to the scientific community worldwide and to publish this paper as is.

---

## Author Response (AR2)

Author Response:

We would like to thank the reviewer for his/her technical comments to improve our manuscript. All the points below have been considered thoroughly and corrected.
Please find our response in red below.

*Sapart et al.,*

Some remaining small issues:

line 62, "The four suggested key mechanisms" — too much certainty! Suggest "Four suggested mechanisms" instead.
This has been corrected.

line 65: Wintereld -> Winterfeld

This has been corrected.
line 71-73: flip the sentence around for clarity: "The remobilized carbon can be used to produce CH4, a strong greenhouse gas (IPCC, 2013) under anaerobic conditions and depending on its type and quality (Schuur et al., 2013)."
("a strong greenhouse gas (IPCC, 2013)" can probably be deleted. I think readers know that...)
This has been corrected.
line 96-97: "However, this fractionation is considered to be relatively small" —> it is relatively small!, so just say "However, this fractionation is relatively small"
This has been corrected.
line 108-109: "The destabilization of gas hydrates is the most discussed CH4 source from this region"
I would disagree with that—there have been many many papers discussing the other sources. Perhaps "The destabilization of gas hydrates is frequently discussed as a CH4 source in this region" is better?
This has been corrected.
line 110: suggest adding reference to Ruppel and Kessler, 2016, The interaction of climate change and methane hydrates, Reviews of Geophysics, 10.1002/2016RG000534.
This has been corrected.
line 115: "radiocarbon content on sediment" —> "radiocarbon content in sediment"
This has been corrected.
line 120: "thus helps determining" —> "thus helps in determining"
This has been corrected.
line 183: "for more detailed on the sample locations" —> "for more detail on the sample locations"
This has been corrected.
line 217: "untypical" —> "atypical"
This has been corrected.
line 243: "with a much more depleted" —> "with much more depleted"

This has been corrected.
line 253 "too depleted" —> "more depleted"
This has been corrected.
line 261: "For these cores and because" —> "For these cores, because"
This has been corrected.
line 262: "on the field" —> "in the field"
This has been corrected.
line 272: " levels below 200pmC thus" —> " levels below 200pmC, thus"
This has been corrected.
line 279: "the sediment showing that it is not originating from the surface" —> Or it could have been pushed down to 30 m from the surface—I don't see how this is disproven in the data. You could write instead "the sediment suggesting that it may not originate from the surface"
This has been corrected.
line 279-280: "Our first assumption is" —> This is not an assumption, this is an informed guess, or a suggested possibility. Suggest: "Our first suggestion is"…
This has been corrected.
line 296: "for a part" —> "partly"
This has been corrected.
line 334-335: "Our dataset does not support this interpretation," —> WRONG. Your dataset says nothing about the "interpretation" of Overduin et al—they were looking at a different core in a different location. Perhaps you mean something like: "Our cores suggest that the Overduin et al core is not typical for the entire area, as we did not observe similar D and 13C enrichments associated with decreases in CH4 concentrations"
This has been corrected.
line 352: "become indiscernible" —> "becomes indiscernible"
This has been corrected.
line 359: "deep Earth's crust" —> "Earth's deep crust"
This has been corrected.
line 373: "for most of it" —> "for the most part"
This has been corrected.
line 418: "at location" —> "at locations"
This has been corrected.
line 424-427: "Our results show that thawing subsea permafrost of the ESAS emits CH4 with an isotopic signature that cannot be easily distinguished from Arctic wetland emissions when looking only at stable isotope data."

This is an important point, but it comes out of nowhere as presented in the conclusions here. No examples of wetland emission dD or d13C are given in the manuscript. Can the authors motivate this earlier in the manuscript? This is the first time the word "wetland" even appears! This isotopic overlap problem is discussed with examples in Thornton et al 2016 "Double-counting challenges the accuracy of high-latitude methane inventories" (GRL) 10.1002/2016GL071772.
A sentence has been added at the end of the introduction to introduce this subject and the Thornton et al., 2016 reference has been added.

line 643: what is the …. in the author list?
This has been corrected.

Supplement

The supplementary material is now much better presented than in the earlier version of this manuscript.

Line 52: "produced at the surface of ice sheets (Baudin et al., 1973)"
There is nothing in the Baudin et al paper about ice sheets (it is about Oklo). Also the year of the Baudin paper is 1972, not 1973. I also did some quick searching for references on 14C production on ice sheets but found nothing.
My apologize, it seems that the references have been mixed up in the revised version of the SI. The production of 14C at the surface of ice sheet is quite well known in the ice core community and several old and recent studies exist about it. I have added here the reference of the study of Firemann and Norris, 1982 who were one of the first to highlight this issue.

Line 54-56: " Nuclear production of 14C involves formation by neutron activation as consequence of a nuclear chain reaction, which may either take place naturally or artificially."

This statement is confusing: neutron activation processes do not require a nuclear chain reaction or a reactor, they only require a source of neutrons. E.g. in the atmosphere cosmic rays produce neutrons which can react with 14N to produce 14C. That said, it is true that radiocarbon can be produced via neutron activation in a reactor, but this is not the only way.

A sentence has been added here: Nuclear production of $^{14}$C involves formation by neutron activation as consequence of a nuclear chain reaction, which may either take place naturally or artificially. In the atmosphere, cosmic rays can also produce neutrons which can react with $^{14}$N to produce $^{14}$C
Line 57: " The only place on Earth, where nuclear fission has occurred naturally"

Incorrect. Oklo is the only place where evidence of a natural nuclear reactor has been found. Nuclear fission occurs naturally at very low rates in U anywhere on the planet, without a reactor.
This has been corrected.
The Baudin et al 1972 reference is very early in studies of Oklo. Some later general overviews, when the Oklo system operations were better understood, include:

Cowan, George A. "A natural fission reactor." Scientific American 235.1 (1976): 36-47.

Kuroda, Paul K. "The Oklo phenomenon." The Origin of the Chemical Elements and the Oklo Phenomenon. Springer Berlin Heidelberg, 1982. 31-55.

These references have been added.